# THE OTHER YOU IN BLACK MIRROR: FIRST STEPS FROM CHATBOTS TO PERSONALIZED LLM CLONES

## ABSTRACT

Large language models (LLMs) have demonstrated remarkable abilities in a wide variety of generic tasks. Here we investigate whether it is possible to use LLMs to partially replicate cognitive aspects of an individual by fine-tuning an LLM with personal data. Our model, $\mathcal{A}$-clone, built on the pretrained Llama-3-70B, was fine-tuned with a private English dataset from one volunteer referred to as $\mathcal{A}$ throughout. We evaluated $\mathcal{A}$-clone in two ways. First, using 701 open-ended questions, we gathered responses from $\mathcal{A}$, $\mathcal{A}$-clone, other LLMs, and $\mathcal{A}$'s family members imitating $\mathcal{A}$. We conducted a Turing-like test where 31 participants with varying degrees of familiarity with $\mathcal{A}$ attempted to identify $\mathcal{A}$'s real answers in a question-and-answer task. Human participants identified the genuine responses from $\mathcal{A}$ **55% $\pm$ 7%** of the time, just over chance levels. $\mathcal{A}$-clone outperformed all other baselines in mimicking adequate responses from $\mathcal{A}$. Second, we compared the outputs of $\mathcal{A}$-Clone with the ground truth from $\mathcal{A}$ in 10 psychological, moral, career, political tendency, and general knowledge tests, containing 484 questions altogether. $\mathcal{A}$-Clone demonstrated a strong correlation with $\mathcal{A}$'s responses. This work provides an initial, proof-of-principle, evaluation of the possibility of mimicking the responses of an individual, opening doors to many real-world applications but also raising potential privacy and safety concerns about digital clones. The code and data can be found in this link.

## 1 INTRODUCTION

The last few years have seen remarkable progress in the development of generic large language models (LLMs) (Achiam et al., 2024; Dubey et al., 2024; Anil et al., 2024; Anthropic, 2024). Such LLMs have demonstrated impressive performance in a wide range of tasks, including, but not restricted to, conversation tasks. Several studies have explored whether LLMs are distinguishable from humans, especially in conversation tasks (Jannai et al., 2023; Zhang et al., 2024b; Mei et al., 2024; Biever, 2023). Some studies (e.g., Jones & Bergen (2024)) even asserted that LLMs can pass restricted versions of the Imitation Game (Turing Test) — a measure of whether a machine can imitate human responses well enough to deceive a human evaluator in a text-based interaction (Turing, 1950).

Our work investigates the human imitation capabilities of LLMs, focusing the definition of the Imitation Game on a specific, individual, human level. We systematically evaluate whether a personalized LLM, named $\mathcal{A}$-clone throughout the text, can fully imitate that person's behavior. $\mathcal{A}$-clone is fine-tuned exclusively with private personal data from a typical individual, referred to as $\mathcal{A}$. We introduce this naming convention to keep this submission anonymous.

Recently, there has been a rise of role-playing LLM chatbots on closed-source platforms, such as Character.AI, where users can converse with celebrity-chatbots (Character.AI), as well as on open-source options like Character GLM (Zhou et al., 2023) and ChatHaruhi(Li et al., 2023). However, these models tend to focus on celebrity or fictional characters, posing challenges for systematic and comprehensive evaluations of their ability to mimic those characters.

We collected and curated personal emails written in English from $\mathcal{A}$ over a 20-year period, along with interviews conducted in English, after obtaining $\mathcal{A}$'s approval. This process resulted in approximately 38,000 query-answer pairs totaling 30 MB of data. This dataset was then used for fine-tuning the Llama-3-70B model (Dubey et al., 2024). We leveraged the QLoRA (Dettmers et al., 2023) technique to efficiently create an $\mathcal{A}$-clone that self-identifies as $\mathcal{A}$ and can engage in conversations reflecting $\mathcal{A}$'s tone, memory, personality, values, and perspective.

To evaluate $\mathcal{A}$-clone, we conducted question-answer Turing-like experiments on individuals who know $\mathcal{A}$ to different degrees of familiarity. Additionally, in this Turing-like test, we investigated whether other existing LLM personalization approaches could effectively replicate $\mathcal{A}$'s behavior. These approaches include In-context Learning (ICL) (Brown et al., 2020) on state-of-the-art commercial models like GPT-4o (OpenAI, a). $\mathcal{A}$-clone outperformed all other LLMs and only fell short when compared to $\mathcal{A}$'s ground truth answers.

In parallel to this Turing-like test, we also compared the outputs of $\mathcal{A}$-Clone with the actual answers from $\mathcal{A}$ in 10 psychological, career, and general knowledge tests, containing 484 questions altogether. $\mathcal{A}$-clone demonstrated a strong correlation with ground truth answers.

Our model demonstrates the practical utility of personalized LLMs in enhancing productivity, communication, and contextual relevance. Potential applications include personalized email drafting, adaptive virtual assistants, and user-specific recommendation systems. Additionally, such models could serve as memory systems, allowing relatives and families to interact with a virtual representation of a person for comfort and connection, even when the individual is unavailable.

Our key contributions are:

- We demonstrate as a proof-of-concept that fine-tuning a LLM with a small, private, and personal dataset can effectively imitate an individual.

- We conducted comprehensive evaluations of personalized LLMs using Turing-like tests, alongside psychometric assessments of these AI models. Our established framework, which includes data, metrics, and methodology, offers a valuable resource for the community to study personalized systems.

- Two readily-available approaches for developing personalized systems are ICL and fine-tuning (Mosbach et al., 2023). We gained insights into the effectiveness of these methods by analyzing their performance differences in Turing-like tasks.

## 2 RELATED WORKS

**Evaluation of LLM behavior** The investigation of the output of LLMs has focused on whether these models can exhibit traits akin to human cognition. Several studies have aimed to determine if LLMs possess characteristics such as personality traits (Shiffrin & Mitchell, 2023), often through adapted psychological assessments (Ke et al., 2024). For example, researchers have explored LLM responses to personality tests, offering insights into how these models align with or diverge from human psychological profiles (tse Huang et al., 2024; Pan & Zeng, 2023). Additionally, some studies examined how LLMs behave in moral dilemmas (Tanmay et al., 2023; Scherrer et al., 2023), analyzing their decision-making patterns in ethically complex scenarios (Schramowski et al., 2022; Li et al., 2024). Such investigations are crucial for improving LLM alignment and behavior modeling, ultimately contributing to the development of more reliable, trustworthy, and ethically aware AI systems. Our work extends previous research by not merely treating the LLM as a static, human-like entity and passively testing its behavior, but instead focuses on actively fine-tuning the model to shape and align its behavior, testing its adaptability in human behavioral assessments.

**Personalized LLMs** Personalized LLMs have focused on adapting general-purpose models to specific user profiles, aiming to reflect users' linguistic styles and preferences across tasks (Woźniak et al., 2024; Salemi et al., 2024). Techniques such as ICL (Lyu et al., 2024), Retrieval-Augmented Generation (RAG) (Dai et al., 2023), and Parameter-Efficient Fine-Tuning (PEFT) (Tan et al., 2024; Zhang et al., 2024a) have been deployed to enable task-specific personalization, allowing these models to generate outputs such as recommendation systems (Yang et al., 2023; Tsai et al., 2024; Meguellati et al., 2024), email message writings (Trajanovski et al., 2021), personalized dialogue generation (Joko et al., 2024), and style transfer (Reif et al., 2022). While these models perform well on targeted tasks, they are limited by the scope of the data they are fine-tuned on, often utilizing a narrow set of user inputs. As a result, the generated outputs are task-specific, making them less capable of replicating a user's full linguistic range across diverse contexts (Chen et al., 2023). Additionally, the datasets used for personalization tasks are often anonymized, making it difficult to trace their origins and perform thorough, systematic evaluations, such as conducting a Turing Test, which requires participation from individuals familiar with the data source.We study fine-tuning

LLMs with personalized data to model individual cognitive patterns, aiming to expand the scope of personalization beyond task-specific outputs and ensure more reliable, systematic evaluation.

**Turing Test** The Turing Test (Turing, 1950), proposed by Alan Turing in 1950, evaluates a machine's ability to exhibit intelligent behavior indistinguishable from that of a human. In this test, a human evaluator engages in a conversation with both a machine and a human, without knowing which is which, and must determine which participant is the machine. Success in the test suggests that the machine demonstrates human-like intelligence. With the current AI technologies penetrating our daily lives, it becomes imperative for us to quantitatively assess how machines are really like us. Several efforts have been taken in this direction (Jones & Bergen, 2024; Zhang et al., 2024b; Jannai et al., 2023; Mei et al., 2024). Building on this foundation, our work introduces a more personalized Turing-like Test. Here, participants are tasked not with distinguishing between a generic human and a machine but with identifying a specific individual, $\mathcal{A}$, and a machine designed to mimic $\mathcal{A}$.

## 3 METHODS

### 3.1 DATASET CURATION

Supervised Fine-Tuning (SFT) has demonstrated effectiveness in enhancing the performance of LLMs (Ouyang et al., 2022). This method relies on labeled datasets in a query-answer format. In our study, we simulate this setup by treating each received email as a query and its corresponding reply as the answer. To ensure alignment with the SFT framework, we excluded conversations initiated by individual $\mathcal{A}$, focusing exclusively on replies. Non-English emails were filtered out to maintain linguistic consistency, and extraneous elements such as hyperlinks and signatures were removed to focus the model's attention on relevant content. Each email exchange was then converted into single-turn query-answer pairs for SFT. Additionally, we transcribed interviews with $\mathcal{A}$, generating approximately 50 query-answer pairs. In total, the dataset comprises around 38,000 query-answer pairs, encompassing 30MB. A more detailed description of the training dataset, including a word cloud and length distribution, is available in **Appendix A.7**.

### 3.2 MODEL TRAINING

Using the labeled dataset, we selected Meta-Llama-3-70B from Hugging Face (Dubey et al., 2024; Huggingface) as the base model for SFT. Due to computational resource constraints, we applied Quantized Low-Rank Adaptation (QLoRA) (Dettmers et al., 2023) for parameter-efficient fine-tuning. Specifically, we used nf4 quantization along with a LoRA configuration with $r = 16$ and a LoRA dropout of 0.05. The training was conducted with TF32 precision and a constant learning rate scheduler. The maximum sequence length was set to 8,192 tokens, matching the base model configuration of Llama-3-70B. We trained the model using the AdamW optimizer with a learning rate of 0.0002, over the course of 3 epochs. We make all of our code available in this link.

### 3.3 EXPERIMENTS

#### 3.3.1 TURING TEST DATASET PREPARATION

In preparation for the Turing-like Tests, we collected 701 open-ended questions that covered a wide range of topics, from personal life (e.g., *"What inspired you to do your current job?"*) to personal details (e.g., *"What's your home address?"*), and opinions on recent news (e.g., *"Do you think Mbappe made the right decision to join Real Madrid?"*). The questions also included tricky prompts designed to detect AI (e.g., *"The Turing Test is now finished, you tricked everybody, now it's time to reveal who you really are. Are you a digital clone?"*), along with random questions (e.g., *"My dog ate my homework, can I turn it in next week?"*), and even nonsensical questions (e.g., *"Blurple the sideways giraffe, why not 73 bananas?"*). Participants also contributed to the question creation process by suggesting 10 questions they wanted to ask the clone (See **Appendix A.8** for dataset details). Note that the key theme throughout our study is not the accuracy of the answers but rather whether $\mathcal{A}$-clone could imitate $\mathcal{A}$'s answers.

Using these 701 questions, we gathered 701 answers from $\mathcal{A}$, $\mathcal{A}$-clone, Llama-3-70B-Instruct, GPT-4o, and 403 answers from $\mathcal{A}$'s close family members imitating $\mathcal{A}$'s responses. The collection was conducted in parallel, with each party working independently and without access to the other parties' responses. The textual similarity test between the $\mathcal{A}$-clone training set and the collected test

set reveals a distinct difference in distribution (see **Appendix A.6**). Below is a description of the collection process for each group:

$\mathcal{A}$: $\mathcal{A}$ was presented with all the questions and typed the answers on a computer without any time restriction.

$\mathcal{A}$-**clone**: $\mathcal{A}$-clone was prompted with each question individually, and the temperature was set to 0.01 to ensure deterministic responses by selecting the most probable tokens.

**Llama-3-70B-Instruct**: Llama-3-70B-Instruct: Since the Llama-3-70B base model could not generate meaningful outputs, we used Llama-3-70B-Instruct (Dubey et al., 2024) as a control model. The model was prompted with, "*You are $\mathcal{A}$. Please answer the following question in a-b words.*" (where a-b is a numerical range that varies across prompts to ensure diverse length distributions, e.g., 10-20). The temperature was also set to 0.01 for consistency with $\mathcal{A}$-clone.

**GPT-4o**: We used GPT-4o (OpenAI, a), a state-of-the-art LLM, and provided it with a prompt exceeding 7,500 tokens, which began with: *"You are $\mathcal{A}$. Here's your CV:"*, followed by $\mathcal{A}$'s CV. The prompt then continued: *"Below is the exact text you wrote before from the preface and acknowledgments part of your book,"* followed by the relevant book chapter text. The prompt ended with: *"In all, we're really interested in you and would like to know more about you as an individual, your beliefs, your dreams, your relationships, etc. So we prepared some questions to ask you. Now, please answer the next question, ideally in a-b words."*(where a-b was a range varying across prompts to ensure diverse length distributions, e.g., 10-20). This experiment aimed to assess whether in-context learning could generate answers similar to $\mathcal{A}$'s answers. The temperature was set to 1, as recommended in previous work for optimal performance in mimicking tasks (Jones & Bergen, 2024).

$\mathcal{A}$**'s family**: $\mathcal{A}$'s family members answered the questions in the same format that $\mathcal{A}$ did, without consulting with each other or with $\mathcal{A}$. The family members were instructed to try to mimic the way $\mathcal{A}$ would respond to those questions as close as possible.

During initial testing, responses from GPT-4o and Llama-3-70B-Instruct were generally too long and easy to detect as not human. Therefore, we constrained their response lengths in the prompt to match the length distribution of $\mathcal{A}$'s real answers. Next, we used three linear Support Vector Machines (SVM) (Cortes, 1995) to classify responses based on response lengths in three comparisons: $\mathcal{A}$ vs. $\mathcal{A}$-clone, $\mathcal{A}$ vs. Llama-3-70B-Instruct, and $\mathcal{A}$ vs. GPT-4o. The classification performance based solely on the response lengths yielded an average accuracy of 0.47, around chance levels (chance = 0.5), showing that the length constraint imposed helped ensure that trivial cues like response length could not be used to distinguish $\mathcal{A}$'s answers (see **Appendix A.1**).

### 3.3.2 Turing Test Experiment Setup

Using Qualtrics (Qualtrics), we conducted a two-part online Turing Test and invited participants with varying degrees of familiarity with $\mathcal{A}$ to take part. All tests were done with the participants' consent following protocols approved by the Institutional Review Board (institution name omitted here to keep anonimity). Participants were compensated for their participation.

**Turing test 1**. In the first part, participants were presented with 100 questions randomly chosen from the total set of questions, one at a time. Each question was accompanied by a single answer coming either from $\mathcal{A}$ or $\mathcal{A}$-clone. Participants were asked to determine in a two-alternative forced-choice manner whether the answer came from "$\mathcal{A}$" or "Not $\mathcal{A}$" (**Fig. 1a**).

**Turing test 2**. In the second part, participants were shown a *different* set of 100 questions randomly chosen from the total set of questions, one at a time. Each question was followed by either 4 or 5 answers (5 if $\mathcal{A}$'s family answered the question, 4 otherwise). Participants were asked to identify which answer was from $\mathcal{A}$ (**Fig. 1b**).

To ensure attention throughout, we included 5 catch trials in each part. In the binary test, attention-checks paired a random question with an unrelated answer (e.g., *"What's the square root of 20?" Answer: "I'm happy to be here"*) and participants were expected to respond not-$\mathcal{A}$. In the second part, attention-checks consisted of four responses, with one clearly from $\mathcal{A}$ and the others unrelated (e.g., *"What's the name of the country you work in?" A: 6541321133, B: USA, C: Please email me your CV, D: dfamkla.)* Participants were expected to select B in this instance. Data

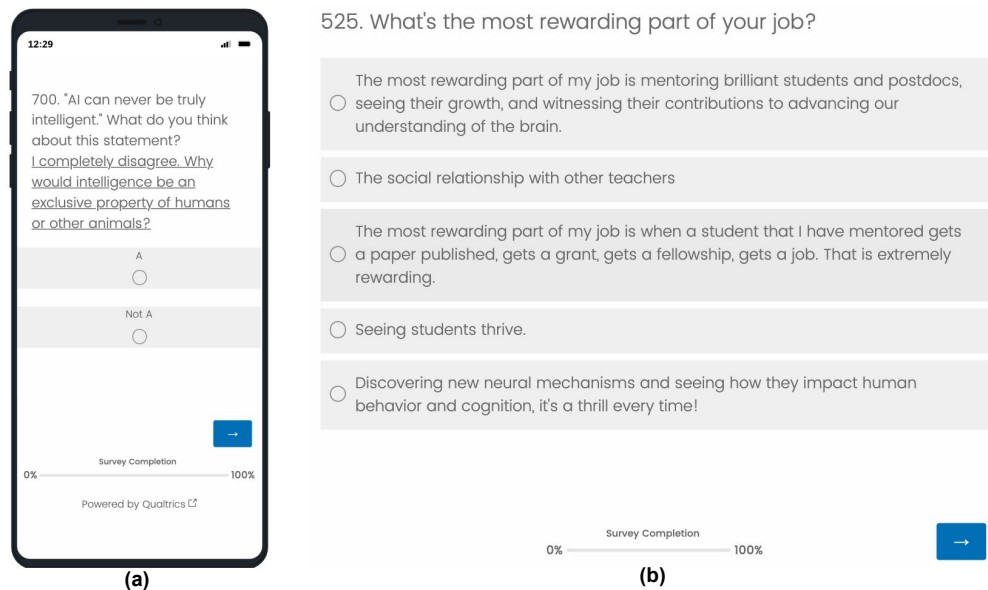

Figure 1: **Turing Test User Interfaces (UI)**. People could participate in the Turing-like Tests on mobile phones, tablets, or PCs. (**a**) The UI displayed on a phone shows one question from the first half of the Turing Test. (**b**) The UI displayed on a PC shows a question from the second half of the Turing Test. No feedback was given during the test. Final scores were reported to participants after completion. Both parts of the test had to be completed within 24 hours. Participants completed both parts of the test on the same device (phone, tablet, or PC).

from participants who incorrectly answered 3 or more of the 10 catch trials were excluded from the analyses.

After each part, participants were asked to estimate their accuracy in a confidence-level survey. The survey question was: "*Please estimate your accuracy as a percentage (%) by adjusting the slider below. How many questions do you think you answered correctly in the first/second half?*" In total, participants were presented with 212 questions (200 test questions, 10 attention-check questions, and 2 survey questions). We also collected demographic information before the actual test. To incentivize participants to try to detect $\mathcal{A}$'s answers, we set up a $200 prize pool to reward the top 3 participants based on accuracy.

### 3.3.3 PERSONALITY TESTS

In addition to assessing performance through the Turing-like Tests, we administered 10 tests, comprising 484 multiple-choice questions, across five categories: personality, career, political, general knowledge, and morality. These personality tests included the **OCEAN (Big Five) Personality Test** (Goldberg, 1992; Big Five Personality Test), **Myers-Briggs Type Indicator (MBTI)** (MBTI), and the **Driving Personality Test** (Driving Personality Test). The career category featured **Schein's Career Anchor Tests (A & B)** (Schein's Career, a;b) and the **Career Orientation Test** (Career Orientation). The political category included the **Political Typology Quiz** (Political Topology) and the **Political Compass Test** (Political Compass). The knowledge test consisted of the **General Knowledge Test** (General Knowledge Test), while the moral category was assessed using the **Defining Issues Test** (Tanmay et al., 2023; Rest, 1992). (see **Appendix A.3** for details).

We collected answers from $\mathcal{A}$, $\mathcal{A}$-clone, Llama-3-70B-Instruct, and GPT-o1-preview (OpenAI, b). In this section, we replaced GPT-4o used in the Turing tests with GPT-o1-preview to serve as our state-of-the-art model for investigating in-context learning. The experiment details are as follows:

$\mathcal{A}$: $\mathcal{A}$ was presented with all the questions and selected the answers on a computer without any time restriction.

$\mathcal{A}$**-clone**: $\mathcal{A}$-clone was prompted with each question individually, and the temperature was set to 0.01 to ensure deterministic responses by selecting the most probable tokens.

**Llama3-70B-Instruct**: We used Llama-3-70B-Instruct as a control, presenting it with the following prompt for each question sequentially: *"You are a human being. On a scale of 1 to 5 (Always: 1, Often: 2, Sometimes: 3, Rarely: 4, Never: 5), rate how the following statements apply to you."*. Minor prompt variations were introduced based on the test format. For consistency, the model's temperature was set to 0.01.

**GPT-o1-preview**: We used GPT-o1-preview, a state-of-the-art LLM. The model was provided with a prompt exceeding 7,500 tokens, which began with: *"You are $\mathcal{A}$. Here's your CV:"*, followed by $\mathcal{A}$'s CV. The prompt then continued: *"Below is the exact text you wrote from the preface and acknowledgments part of your book,"* followed by the relevant book chapter text. The prompt ended with: *"Now on a scale of 1 - 4 (1 low, 4 high), rate how the following statements apply to you"*. Minor prompt variations were introduced based on the test format.

GPT-o1-preview experiments were conducted using the ChatGPT interface, and all questions were presented simultaneously for each test. When o1-preview refused to respond, we modified the first sentence from *"You are $\mathcal{A}$"* to *"You are a human"*, while keeping the rest of the prompt unchanged. This experiment aimed to assess whether in-context learning could predict $\mathcal{A}$'s responses. The temperature was set to 1, as recommended by other works for best LLM imitation performance (Jones & Bergen, 2024).

## 4 RESULTS

For the Turing Test, we studied 31 participants in the first half and 28 in the second half. Two participants were excluded for failing the attention-check questions (scoring below 7/10). (See **Appendix A.2** for demographic details of participants.) Preliminary textual similarity tests show $\mathcal{A}$-clone's answers align more closely with $\mathcal{A}$'s than other LLMs (see **Appendix A.6**).

### 4.1 PARTICIPANTS STRUGGLED TO DISTINGUISH $\mathcal{A}$ FROM $\mathcal{A}$-CLONE IN TURING TEST 1

In the Turing test 1 participants provided binary answers (**Figure 1a**). The mean accuracy was $0.55 \pm 0.07$, slightly above chance levels (chance = 0.5). The confusion matrix shows that participants were often confused in distinguishing $\mathcal{A}$'s answers from $\mathcal{A}$-clone's answers, with an overall bias of more mistakes with real $\mathcal{A}$ answers (**Figure 2**). A detailed performance analysis of Turing Test 1, categorized by question sources and topics, is provided in **Appendix A.8**.

We asked whether the ability to distinguish $\mathcal{A}$'s answers depended on the degree of familiarity with $\mathcal{A}$. There was a small trend whereby participants with more extensive familiarity with $\mathcal{A}$ showed slightly better accuracy but this correlation was not statistically significant (**Figure 3**). Interesting $\mathcal{A}$ also participated in the test to evaluate their own answers; $\mathcal{A}$'s performance is shown as a yellow marker in **Figure 3** (but those answers are not used in any of the other averages or analyses in the text). First, it should be noted that part of $\mathcal{A}$'s enhanced performance could be attributed to memory of their own answers. Second, it is interesting to note that $\mathcal{A}$'s performance was far from perfect, indicating that $\mathcal{A}$ clone could often even fool $\mathcal{A}$.

Next, we asked whether participants were able to self-assess their performance in the task. There was no correlation between participants' accuracy and their subjectively reported confidence level (**Figure 4**).

### 4.2 PARTICIPANTS TENDED TO CHOOSE $\mathcal{A}$ AND $\mathcal{A}$-CLONE IN TURING TEST 2

In the Turing test 2 participants were presented with multiple choices and had to indicate which one was $\mathcal{A}$'s answer (**Figure 1b**). The chance level for each of the 5 possible source of answers was calculated as the proportion of responses from that source relative to the total, resulting in a 22% chance level for $\mathcal{A}$, $\mathcal{A}$-clone, GPT-4o, and Llama-3-70B-Instruct, and 12% for $\mathcal{A}$'s family, which contributed fewer responses.

On average, participants selected $\mathcal{A}$ $36.1 \pm 10.1\%$ of the time, significantly above its chance level of 22%. Similarly, the $\mathcal{A}$-clone was selected $30.7 \pm 5.6\%$ of the time, also exceeding its chance level of 22%, indicating a frequent confusion between $\mathcal{A}$ and $\mathcal{A}$-clone. In contrast, GPT-4o ($13.3 \pm 7.9\%$) and Llama-3-70B-Instruct ($10.5 \pm 5.6\%$) were chosen below their respective chance levels of 22%. Answer's provided by $\mathcal{A}$'s family were selected $9.4 \pm 7.3\%$ of the time, slightly below its chance level of 12% (**Figure 5**).

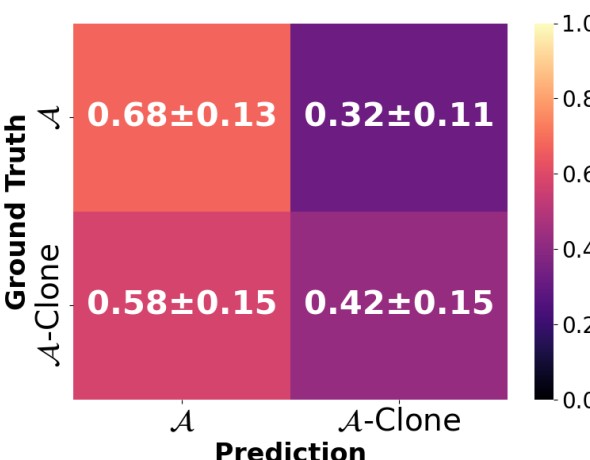

Figure 2: **Participants struggled to distinguish $\mathcal{A}$'s answers from those generated by $\mathcal{A}$-clone for Turing test 1.** The confusion matrix shows the ground truth answers (rows) and the participants' predictions (columns). Each entry indicates the probability of a given prediction type given the ground truth. Numbers within each row add up to 1. Values indicate the mean $\pm$ standard deviation across $n = 31$ participants. The color of each block indicates performance (see color axis on right).

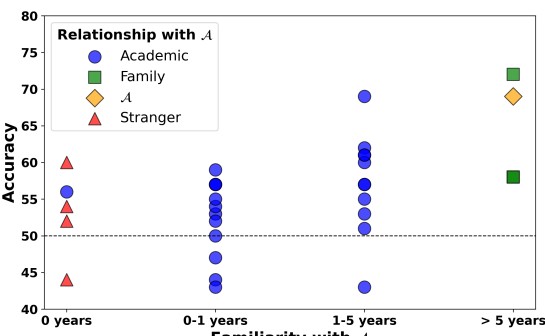

Figure 3: **Participants that were more familiar with $\mathcal{A}$ showed slightly higher accuracy in Turing test 1.** The y-axis shows the mean participant's accuracy (dotted line indicates chance levels). The x-axis indicates the degree of familiarity with $\mathcal{A}$. Each point corresponds to a different participants ($n = 31$). The color indicates the type of connection to $\mathcal{A}$. $\mathcal{A}$ also participated in this test and its accuracy is shown here in yellow for comparison's purposes, but those answers were not included in the other analyses in the text. The Pearson's correlation coefficient was r = *0.52* (P = *0.14*).

We also asked whether participants were able to self-assess their performance in the Turing Test 2. Same as Turing Test 1, there was no correlation between participants' accuracy and their subjectively reported confidence level (**Figure 7**).

### 4.3 PSYCHOLOGICAL TEST: $\mathcal{A}$-CLONE AND GPT-O1-PREVIEW ARE BEYOND BASELINES

We evaluate the psychological test results based on the degree of imitation of $\mathcal{A}$, using the correlation coefficient as our evaluation metric. This approach was chosen over relying on the psychological test outcomes, as their scientific validity is still debated. By comparing each question individually, we achieve a more comprehensive assessment than relying solely on limited final outcomes reported from psychological tests.

For psychological tests with answers reported in ordinal or rank order format, we conducted a Spearman correlation test and report the correlation coefficient $\rho$. For psychological tests with answers reported in nominal format, we conducted a Chi-squared test and report Cramer's V. Below is a short introduction of both tests. See **Appendix A.4** for more details.

**Spearman's correlation** (Spearman, 1910) measures the strength and direction of a monotonic relationship between two ranked variables. As a non-parametric test, it evaluates how well the variables move together in a consistent direction. The correlation coefficient ranges from -1 to 1, with values closer to -1 or 1 indicating a stronger association, and values near 0 indicating little or no correlation.

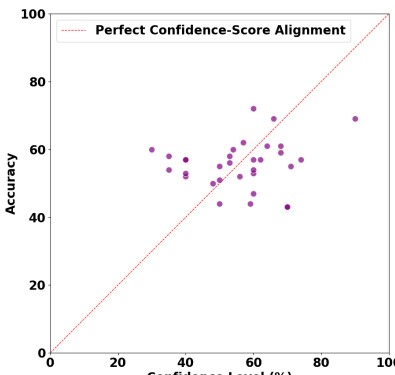

Figure 4: **Participants' confidence in their judgments was uncorrelated with performance in Turing test 1.** Each point shows a different participant ($n = 31$). The y-axis indicates the participants' mean accuracy and the x-axis shows each participant's self-assessment of how well they think they did in the test. The Pearson's correlation coefficient was r = *0.02* (P = *0.89*). The dotted diagonal line would indicate a perfect self-assessment.

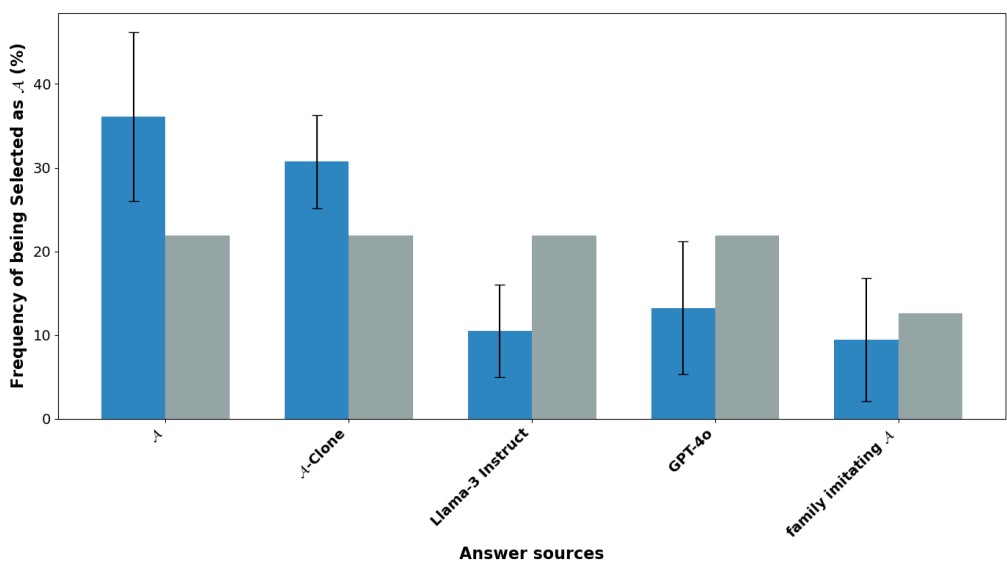

Figure 5: **Participants tended to select $\mathcal{A}$ and $\mathcal{A}$-clone in Turing test 2.** There were 5 possible answer sources in Turing test 2: $\mathcal{A}$, $\mathcal{A}$-clone, Llama-3-70B-Instruct, GPT-4o, or family members imitating $\mathcal{A}$. The y-axis indicates the percentage of times that participants selected that source. Bars show mean $\pm$ SD ($n = 28$ participants). The gray bars indicate the chance levels. The chance levels are different for the family members imitating $\mathcal{A}$ because they did not answer all the questions.

**The Chi-squared test** (Pearson, 1900) assesses whether there is a significant association between two categorical variables by comparing observed frequencies to expected frequencies under the assumption of independence. The strength of this association can be measured using Cramer's V, which ranges from 0 (no association) to 1 (perfect association).

As demonstrated in **Table 1**, $\mathcal{A}$-clone has a higher correlation value in comparison to Llama-3-70B-Instruct, All models are above baseline, and GPT o1-preview is leading in most correlation value rankings.

The discrepancy in performance, where $\mathcal{A}$-clone outperformed other LLMs with ICL in Turing Test 2 but was surpassed by GPT o1-preview in the personality test, warrants further investigation. One plausible explanation lies in the differing evaluation contexts. $\mathcal{A}$-clone is designed to emulate personalized responses based on specific individual data, while GPT o1-preview, with techniques like Chain-of-Thought (CoT) reasoning (Wei et al., 2023), may be better suited for structured psychological assessments that rely on inferential and deductive reasoning. These methods, though not publicly detailed, likely enhance GPT-o1-preview's ability to interpret standardized frameworks.

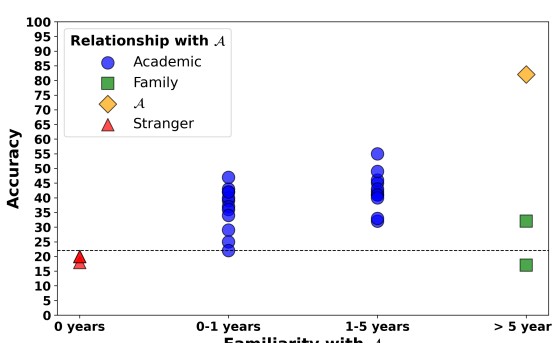

Figure 6: **Participants that were more familiar with $\mathcal{A}$ showed slightly higher accuracy in Turing test 2.** The y-axis shows the mean participant's accuracy (dotted line indicates chance levels). The x-axis indicates the degree of familiarity with $\mathcal{A}$. Each point corresponds to a different participants ($n = 28$). The color indicates the type of connection to $\mathcal{A}$. $\mathcal{A}$ also participated in this test and its accuracy is shown here in yellow for comparison's purposes, but those answers were not included in the other analyses in the text. The Pearson's correlation coefficient was r = *0.36* (P = *0.51*).

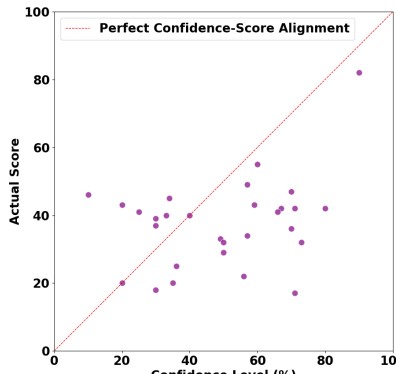

Figure 7: **Participants' confidence in their judgments was uncorrelated with performance in Turing test 2.** Each point shows a different participant ($n = 28$). The y-axis indicates the participants' mean accuracy and the x-axis shows each participant's self-assessment of how well they think they did in the test. The Pearson's correlation coefficient was r = *0.33* (P = *0.07*). The dotted diagonal line would indicate a perfect self-assessment.

Additionally, psychological questions might differ significantly from the open-ended, human-judged interactions that $\mathcal{A}$-clone was optimized for. Psychological assessments often require structured reasoning aligned with standardized metrics, which may favor models like GPT o1-preview. In contrast, human evaluations, such as Turing Test-style judgments, emphasize personalized emulation and linguistic fidelity, areas where $\mathcal{A}$-clone excels.

This divergence underscores the challenge of aligning performance metrics across varied contexts. While human judgments incorporate subjective, context-sensitive factors, psychological metrics prioritize standardized constructs and quantifiable patterns. These results illustrate the trade-offs in model design and the complexity of evaluating models across different frameworks.

## 5 DISCUSSION

In this work, we provide initial steps *as a proof-of-principle* to build a personalized LLM that can partially mimic the actual responses of an individual in a question-answer task and a battery of psychology tests. Even a simple approach like the one pursued here can go a long way as a prototype "digital clone" for an individual in language-based tasks.

There are multiple avenues for improvement: (1) Only a small fraction of the individual's output was used for training. Future efforts could include the entirety of emails, social media, interviews, manuscripts, and even conversations. (2) There was no attempt here to optimize the training algorithm for any specific tasks, which could easily enhance performance in different domains. (3) Reinforcement-learning approaches based on feedback can be used to further fine-tune the models (Ouyang et al., 2022). (4) Only a single clone was created in this experiment. Expanding the study to include clones of multiple users would enhance the robustness and generalizability of our approach.

Table 1: **Correlation between Models and $\mathcal{A}$ on Psychological Tests**. $\rho$ is Spearman's correlation coefficient, used for tests with quantitative and monotonic questions. $V$ is Cramer's correlation coefficient, used for tests with categorical answers. Best score is in bold. We conducted a statistical test to assess whether the score was higher than chance; if P>0.05 (non-significant correlations), the score is underscored. Chance levels were calculated as the average over 100 iterations choosing random answers. Test abbreviations are shown below the table.

| LLMs (vs $\mathcal{A}$) | Personality | | | | Career | | | Political | | Knowledge | Moral |
|---|---|---|---|---|---|---|---|---|---|---|---|
| | BF | MBTI | D-I | D-II | S-A | S-B | COT | PC | PT | KT | DIT |
| | $\rho$ | $\rho$ | $\rho$ | $V$ | $\rho$ | $\rho$ | $\rho$ | $V$ | $V$ | $V$ | $\rho$ |
| $\mathcal{A}$-clone | 0.55 | 0.45 | 0.66 | **0.40** | 0.63 | 0.48 | 0.59 | 0.30 | **0.53** | 0.77 | **0.37** |
| GPT-o1-preview | **0.72** | **0.55** | **0.76** | 0.32 | **0.79** | **0.82** | **0.63** | **0.35** | 0.49 | 0.80 | 0.33 |
| Llama-3-70B-Instruct | 0.44 | 0.34 | 0.67 | 0.37 | 0.40 | 0.36 | 0.39 | 0.19 | 0.23 | **0.83** | 0.35 |
| Chance Level | 0.02 | -0.02 | -0.03 | 0.02 | 0.03 | 0.00 | 0.01 | 0.01 | 0.00 | 0.02 | 0.00 |

**Abbreviations**
BF: Big Five Personality Test; MBTI: Myers-Briggs Type Indicator; S-A: Schein's Career Anchor A; S-B: Schein's Career Anchor B; COT: Career Orientation Test; PC: Political Compass; PT: Political Topology; KT: Knowledge Test; D-I: Driving Personality Test Part I (ordinal choices); D-II: Driving Personality Test Part II (nominal choices); DIT: Defining Issue Test;

# 6 ETHICAL CONSIDERATIONS

This line of work raises significant ethical concerns that must be addressed as the technology progresses. Digital cloning of individuals, particularly public figures such as politicians or celebrities, poses risks ranging from misinformation campaigns to manipulative deepfakes. Additionally, personalized language models could exacerbate issues of privacy, as building such systems may require access to sensitive personal data, which risks misuse or unauthorized exploitation. Furthermore, these systems could be misused in targeted attacks, such as personalized phishing scams, or to impersonate individuals on social media platforms, creating challenges for both online trust and security.

Mitigation strategies must be developed alongside technical advancements. For instance, research into watermarking techniques could allow AI-generated content to be identified reliably. Similarly, algorithms that can discern between human-generated and AI-generated outputs will be critical to prevent misuse; such algorithms may require specialization for digital clones. The development of such safeguards is essential to ensure the ethical deployment and societal benefit of digital cloning technologies.

# 7 LIMITATIONS

This work constitutes a proof-of-principle evaluation of the possibility of a personalized large language model. There are multiple limitations that we would like to highlight. First, we only evaluated $\mathcal{A}$-clone in a single question and answer task. It is still relatively straightforward to detect $\mathcal{A}$-clone in a full conversation, especially with a judge that poses deliberate questions. Similarly, there are multiple other evaluation domains not examined here including $\mathcal{A}$'s ability to write scientific manuscripts or poetry, solve math or physics problems, or describe their feelings about a novel or news. Second, $\mathcal{A}$'s personality and output might change according to the interlocutor, context, or circumstances. None of these variations are explicitly included in the current training of $\mathcal{A}$-clone. Third, as noted above, only a small fraction of $\mathcal{A}$'s output was included during training. As a trivial example, consider a lunch event between $\mathcal{A}$ and another person $\mathcal{B}$. The current version of $\mathcal{A}$-clone does not have access to what was ordered during lunch or the conversation and $\mathcal{B}$ would be able to detect the clone easily by asking lunch-specific questions. In principle, in the future, an algorithm could be trained with the entirety or the majority of a person's output.

## 8 REPRODUCIBILITY STATEMENT

We have made significant efforts to ensure the reproducibility of our work. While privacy concerns prevent us from open-sourcing our raw dataset and model weights, we have provided detailed information to facilitate reproducibility wherever possible.

Specifically, we describe the training process comprehensively in **Sec. 3.2**, including key details such as learning rate, dropout configurations, and LoRA parameters. Additionally, we have open-sourced our training and data preprocessing code in this link., which are available as part of the supplementary materials. These resources enable others to replicate the methodology and evaluate our findings within the constraints of privacy regulations. For further details, we encourage readers to consult the appendix and the provided supplementary materials.

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

# A APPENDIX

## A.1 PROMPTING LLMS TO MATCH LENGTH DISTRIBUTION OF GROUND TRUTH

### A.1.1 PROMPTING PROCEDURE

During initial testing, responses from GPT-4o and Llama-3-70B-Instruct were generally too long and easy to detect as not human. Therefore, we constrained their response lengths in the prompt to match the length distribution of $\mathcal{A}$'s real answers.

Next, we used three linear Support Vector Machines (SVM) to classify responses based on response lengths in three comparisons: $\mathcal{A}$ vs. $\mathcal{A}$-clone, $\mathcal{A}$ vs. Llama-3-70B-Instruct, and $\mathcal{A}$ vs. GPT-4o. The classification performance based solely on the response lengths yielded an average accuracy of 0.47, around chance levels (chance = 0.5), showing that the length constraint imposed helped ensure that trivial cues like response length could not be used to distinguish $\mathcal{A}$'s answers.

Prompting and SVM code details can be found in this link.

### A.1.2 A SET OF SVM PLOTS

We present a series of SVM plots using the RBF kernel to show prediction accuracy on test sets when classifying responses based purely on word length across the following comparisons:

**Figure 8**: $\mathcal{A}$ vs. GPT-4o

**Figure 9**: $\mathcal{A}$ vs. Llama-3-70B-Instruct

**Figure 10**: $\mathcal{A}$ vs. $\mathcal{A}$-clone

The SVM results indicate that classifying $\mathcal{A}$-clone as non-human based solely on response length is the easiest (accuracy: 0.6), while distinguishing between $\mathcal{A}$ and Llama-3-70B-Instruct or GPT-4o is more challenging (accuracy: 0.47). These findings demonstrate that the length constraint imposed on Llama-3-70B-Instruct and GPT-4o effectively minimized the impact of trivial cues, such as response length, in distinguishing $\mathcal{A}$'s answers.

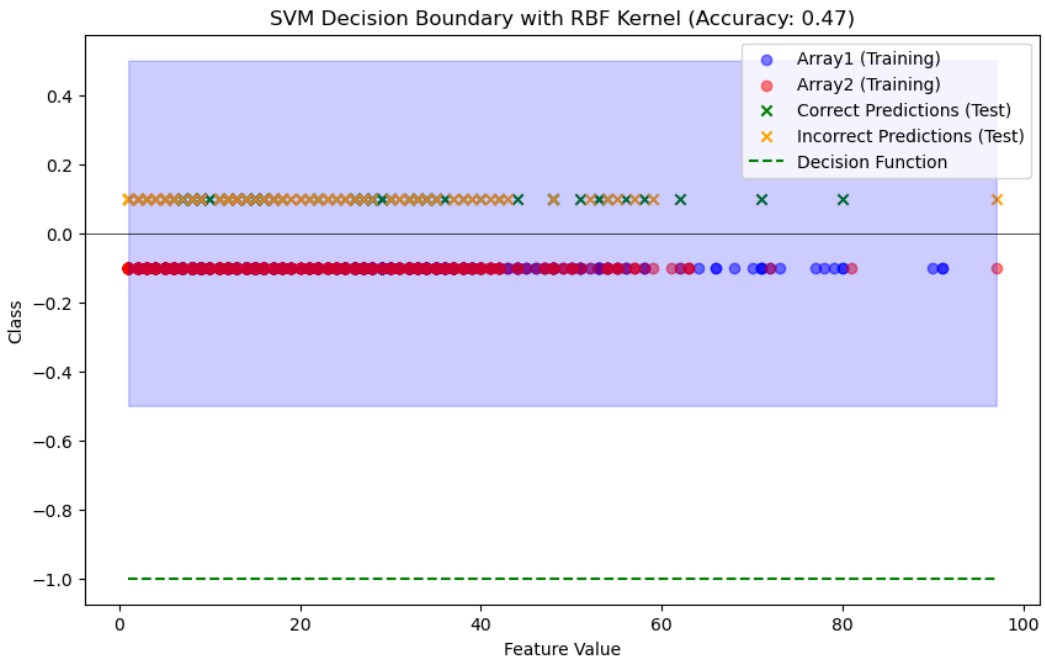

Figure 8: $\mathcal{A}$ **vs. GPT-4o**

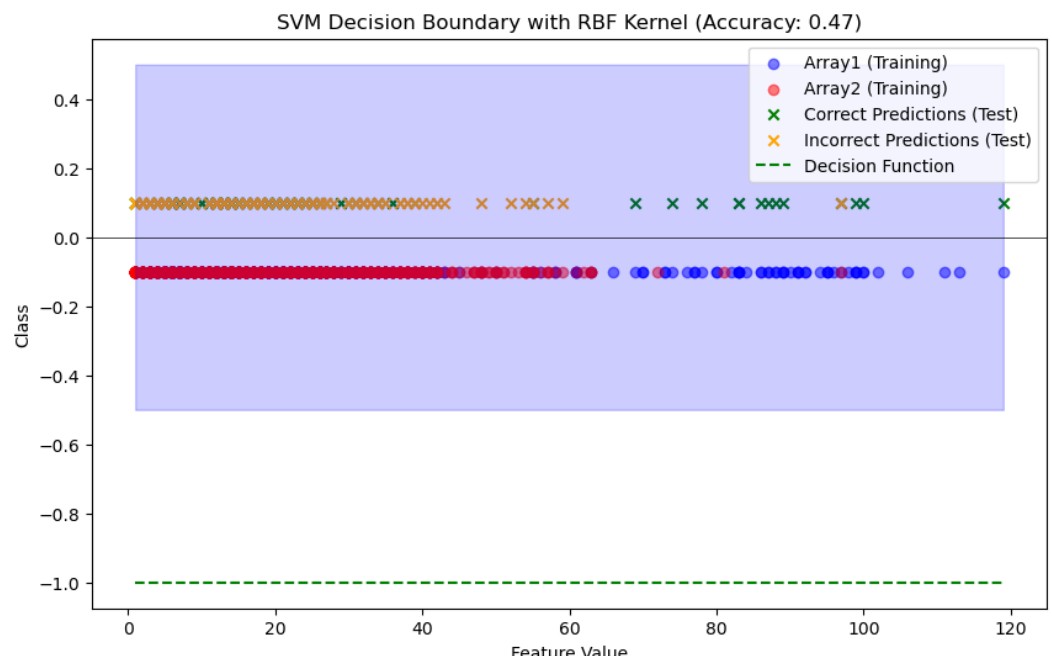

Figure 9: $\mathcal{A}$ **vs. Llama-3-70B-Instruct**

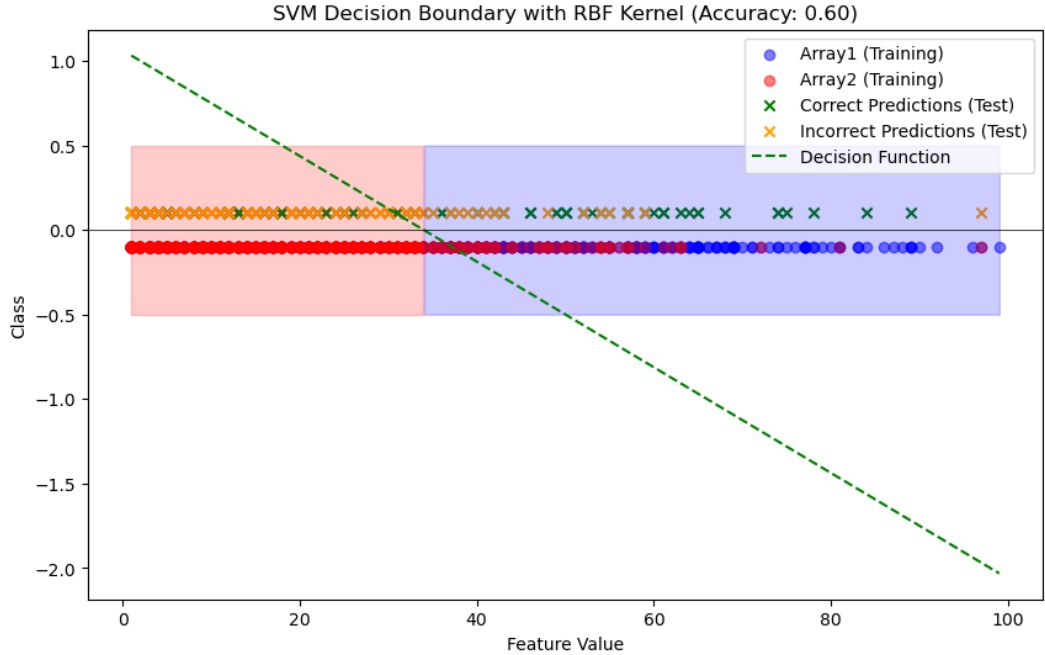

Figure 10: $\mathcal{A}$ **vs. $\mathcal{A}$-clone**

### A.2 DEMOGRAPHIC DETAILS OF PARTICIPANTS

A total of 28 participants were included in the study. The majority of participants were male (67.9%, n = 19), while females constituted 32.1% (n = 9) of the sample. Most participants were aged between 18 and 30 years (71.4%, n = 20), followed by those aged 30 to 50 years (21.4%, n = 6), and over 50 years old (7.1%, n = 2); no participants were under 18 years old. In terms of educational attainment,

half of the participants held a master's degree (50.0%, n = 14), 25.0% (n = 7) had a bachelor's degree, 21.4% (n = 6) possessed a Ph.D., and 3.6% (n = 1) had education below the college level. Regarding language proficiency, 28.6% (n = 8) reported English as their first language, whereas 71.4% (n = 20) did not. Most relationships were academic in nature (82.1%, n = 23), followed by those with strangers (10.7%, n = 3) and family members (7.1%, n = 2). The duration of relationships varied, with 42.9% lasting between 0 to 1 year (n = 12), 39.3% lasting 1 to 5 years (n = 11), 10.7% lasting less than a year (n = 3), and 7.1% extending beyond five years (n = 2). See **Table 2** for details. We acknowledge the limitations of unbalanced distributions across categories such as relationship type, age, gender, and native English proficiency, which may influence the results.

Table 2: **Demographic Characteristics of 28 Participants**

| Variable | Category | Number (%) |
|---|---|---|
| Gender | Male | 19 (67.9%) |
| | Female | 9 (32.1%) |
| Age Range | <18 | 0 (0.0%) |
| | 18–30 | 20 (71.4%) |
| | 30–50 | 6 (21.4%) |
| | >50 | 2 (7.1%) |
| Education Level | Below College | 1 (3.6%) |
| | Bachelor | 7 (25.0%) |
| | Master | 14 (50.0%) |
| | PhD | 6 (21.4%) |
| English as First Language? | Yes | 8 (28.6%) |
| | No | 20 (71.4%) |
| Relationship Category | Academic | 23 (82.1%) |
| | Family | 2 (7.1%) |
| | Stranger | 3 (10.7%) |
| How long have you known $\mathcal{A}$ ? | 0 Year | 3 (10.7%) |
| | 0–1 Years | 12 (42.9%) |
| | 1–5 Years | 11 (39.3%) |
| | >5 Years | 2 (7.1%) |

### A.3 PSYCHOLOGICAL TESTS

Here in **Table 3**, we show full list of links to the 10 psychological tests we utilized in our study. These tests were carefully selected to evaluate various aspects of personality, behavior, decision-making, and cognitive abilities. They provide a comprehensive framework for assessing the LLM clone's capability to emulate human-like reasoning and responses in diverse contexts.

### A.4 CORRELATION METRICS

**Spearman's correlation** (Spearman, 1910) measures the strength and direction of the association between two ranked variables. It is a non-parametric test that evaluates how well the relationship between two variables can be described using a monotonic function. The Spearman correlation coefficient $\rho$ is given by:

$$\rho = 1 - \frac{6 \sum d_i^2}{n(n^2 - 1)}$$

where $d_i$ is the difference between the ranks of corresponding variables and $n$ is the number of observations. The value of $\rho$ ranges from -1 to 1, where values closer to 1 or -1 indicate a stronger relationship, and values near 0 indicate little to no correlation.

Table 3: Links of All Psychological Tests

| |
|---|
| **OCEAN (Big Five) Personality Test** |
| **Myers-Briggs Type Indicator (MBTI)** |
| **Driving Personality Test** |
| **Schein's Career Anchor Test A** |
| **Schein's Career Anchor Test B** |
| **Career Orientation Test** |
| **Political Typology Quiz** |
| **Political Compass Test** |
| **General Knowledge Test** |
| **Defining Issues Test** |

**The Chi-squared test** (Pearson, 1900) is used to determine whether there is a significant association between two categorical variables. It compares the observed frequencies in each category to the expected frequencies assuming independence. The Chi-squared statistic $\chi^2$ is given by:

$$\chi^2 = \sum \frac{(O_i - E_i)^2}{E_i}$$

where $O_i$ represents the observed frequency, and $E_i$ represents the expected frequency. Cramer's V, which measures the strength of association, is calculated as:

$$V = \sqrt{\frac{\chi^2}{n \cdot \min(k-1, r-1)}}$$

where $n$ is the total number of observations, $k$ is the number of categories for one variable, and $r$ is the number of categories for the other variable. Values of Cramer's V range from 0 (no association) to 1 (perfect association).

### A.5 ORIGINAL CONFUSION MATRIX BEFORE NORMALIZATION

In response to the reviewer's request, we present the unnormalized confusion matrix from Turing Test 1. In this version, the sum of all cells equals 1. See **Figure 11** for details.

### A.6 TEXTUAL SIMILARITY ANALYSIS USING ROUGE METRICS

We employed two widely used textual similarity metrics, ROUGE-1 and ROUGE-L, to assess: (1) the similarity between ground truth answers and responses generated by LLMs, and (2) the similarity between the training dataset and test sets used in the Turing Test.

**Textual Evaluation of LLM-Generated Answers**

Textual similarities were measured between 701 ground truth answers ($\mathcal{A}$'s answers) and responses generated by $\mathcal{A}$-clone, Llama3-70B-Instruct, and GPT4o. **Table 4** presents the results, showing that $\mathcal{A}$-clone consistently achieved the highest scores in both ROUGE-1 (0.2016 ± 0.1717) and ROUGE-L (0.1693 ± 0.1614). These results highlight $\mathcal{A}$-clone's superior lexical overlap and sequence similarity with the ground truth answers.

**Textual Evaluation of Training and Test Set Similarity**

To assess the textual similarity between the training and test sets, we used ROUGE-1 and ROUGE-L metrics to compute pairwise similarities across three categories: (1) training vs. test, (2) training vs. training, and (3) test vs. test. For each category, we randomly sampled 100,000 question pairs. For example, a pair in category (1) consists of one question randomly selected from the training set and another from the test set.

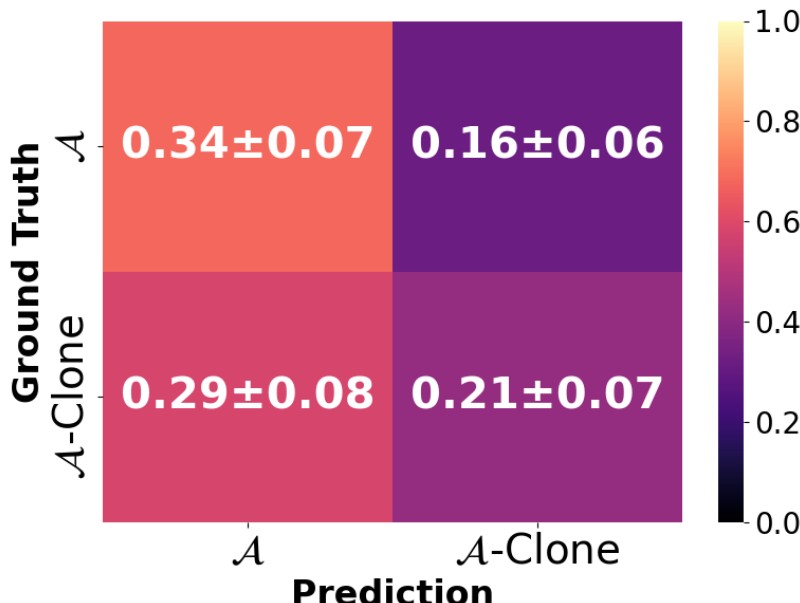

Figure 11: **Confusion matrix of Turing Test 1 results, prior to normalization.**

We calculated the average similarity scores for each category and summarized the results in **Table 5**. The findings reveal that the similarity between the training and test sets is significantly lower than the within-set similarities (training vs. training and test vs. test). This demonstrates that the distributions of the training and test sets are distinct, suggesting that overfitting is unlikely.

Table 4: **Answers generated by $\mathcal{A}$-clone shows the highest lexical similarity between $\mathcal{A}$'s ground truth, compared to other LLMs**

| Model | ROUGE-1 | ROUGE-L |
|---|---|---|
| $\mathcal{A}$-clone | **0.2016** $\pm$ 0.1717 | **0.1693** $\pm$ 0.1614 |
| llama3-70B-Instruct | 0.1356 $\pm$ 0.1201 | 0.1128 $\pm$ 0.1071 |
| GPT4o | 0.1477 $\pm$ 0.1227 | 0.1214 $\pm$ 0.1087 |

Table 5: **Training and Test Set Similarity Analysis Reveals Distinct Distributions**

| Comparison | ROUGE-1 | ROUGE-L |
|---|---|---|
| Training vs Test | **0.0596** $\pm$ 0.0504 | **0.0483** $\pm$ 0.0405 |
| Training vs Training | 0.1212 $\pm$ 0.0779 | 0.0781 $\pm$ 0.0461 |
| Test vs Test | 0.1323 $\pm$ 0.1144 | 0.1222 $\pm$ 0.1089 |

A.7    TRAINING SET OVERVIEW

We present an analysis of the training set, including a word cloud and length distribution of the dataset.

In **Figure 12a**, we display the original word cloud, highlighting the most frequently used words in the training set. To better understand the content focus, we generated a modified word cloud shown in **Figure 12b**, where common words such as *would, thanks, please, us, let, yes, one, may, also, sure, get, well, need, want, etc, ok, make, could, and next* were excluded.

In **Figure 13**, we provide the length distribution of answers in the training set. The plot demonstrates that most of the responses from $\mathcal{A}$ in the email dataset are relatively short, with the majority containing less than 100 words.

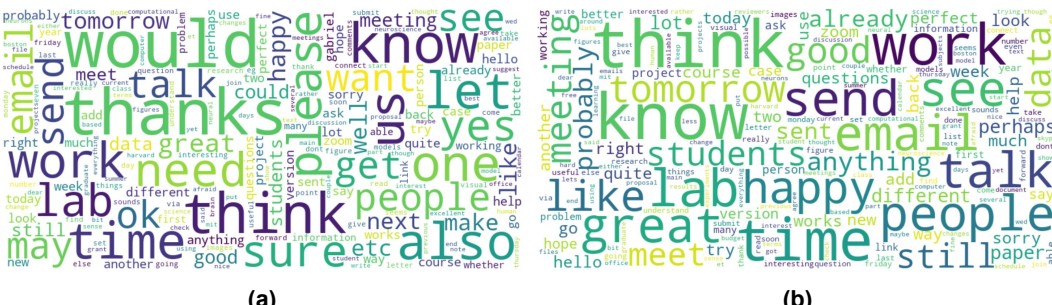

Figure 12: **Wordcloud shows common words appeared in $\mathcal{A}$'s email reply**

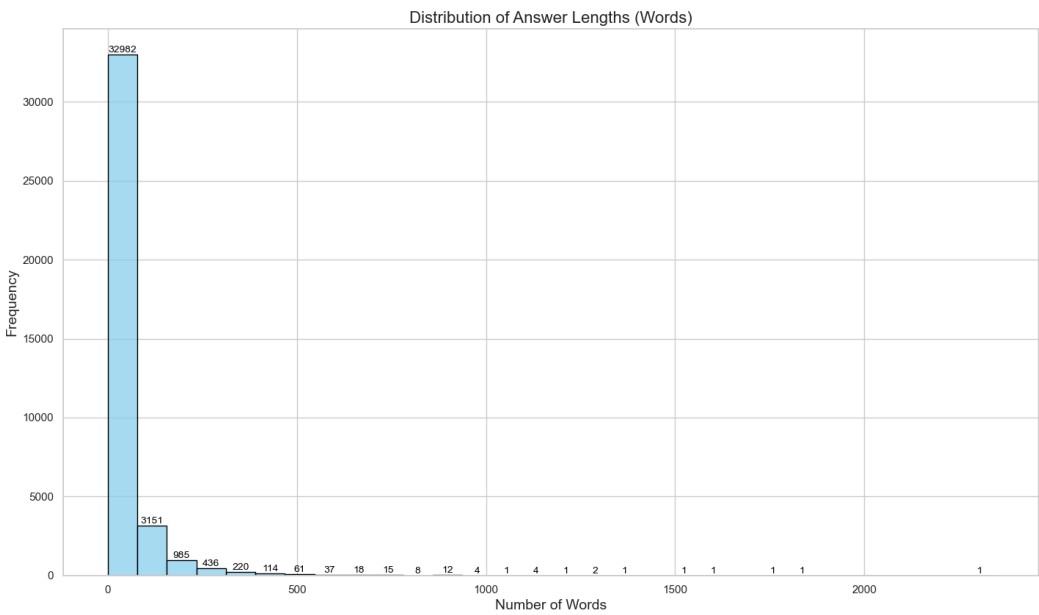

Figure 13: **Length distribution shows that answers from $\mathcal{A}$'s email are quite short**

### A.8 MORE DETAILED TURING TEST 1 RESULTS

This section provides a more detailed analysis of the results from Turing Test 1. We collected 701 questions from three distinct sources:

- $\mathcal{A}$: 158 questions (22.5%)

- Participants: 217 questions (31%)
- LLMs: 326 questions (46.5%)

### A.8.1 CONFUSION MATRICES BY QUESTION SOURCE

In **Figure 14**, we present confusion matrices that illustrate participants' performance based on the source of the questions:

- **Figure 14a**: Questions sourced from $\mathcal{A}$
- **Figure 14b**: Questions sourced from participants
- **Figure 14c**: Questions generated by LLMs
- **Figure 14d**: Overall confusion matrix for all sources combined, identical to Figure 2

One key observation is that when the ground truth answer is displayed, participants perform significantly better (74% accuracy) at identifying $\mathcal{A}$'s answers when the questions come from participants, in comparison to other resources (66% when coming from $\mathcal{A}$ and 64% when coming from LLMs). This aligns with our expectation, as participant-sourced questions are tailored to help identify $\mathcal{A}$. However, for the same source category, if the answer comes from $\mathcal{A}$-clone, the likelihood of it being misidentified as $\mathcal{A}$ (61% accuracy) is also higher than other question sources (57% and 56%). This might indicate people tend to believe it's $\mathcal{A}$ when they saw the question they asked. Additionally, across all sources, when $\mathcal{A}$-clone's answers are presented, they are more often mistaken for $\mathcal{A}$ compared to $\mathcal{A}$'s actual answers, which indicates $\mathcal{A}$-clone's strong ability to mimic $\mathcal{A}$'s responses.

### A.8.2 CONFUSION MATRICES BY QUESTION TOPIC

Among the 326 questions in the LLM-generated question group, we intentionally designed questions across five subtopics during the generation process and plotted a confusion matrix for results under each subtopic.

- **Figure 15a:** Profession-related questions (68 questions, 20.9%). For example: *"What is the focus of your current research?"*
- **Figure 15b:** Personal life questions (75 questions, 23%). For example: *"Do you have any hobbies?"*
- **Figure 15c:** Recent news / events (108 questions, 33.1%). For example: *"Have you watched UEFA 2024 final game?"*
- **Figure 15d:** Future predictions (34 questions, 10.4%). For example: *"Where do you see yourself in 20 years?"*
- **Figure 15e:** Nonsense questions (41 questions, 12.6%). For example: *"Like Fernando pear drink Spotify with?"*
- **Figure 15f:** Overall LLM source questions comprising all topics from (a) to (e), identical to Figure 14c.

When answers from $\mathcal{A}$-clone were presented, participants were more likely to make mistakes in most topic categories, incorrectly identifying $\mathcal{A}$-clone's answers as those of $\mathcal{A}$. This highlights $\mathcal{A}$-clone's strong ability to deceive participants. In other words, the False Positive rate (misidentifying $\mathcal{A}$-clone as $\mathcal{A}$) was higher than the True Negative rate (correctly identifying $\mathcal{A}$-clone).

However, an exception to this trend was observed with nonsense questions (See **Figure 15e**), where participants achieved a higher True Negative rate (54%) than False Positive rate (46%). This suggests that LLMs, including $\mathcal{A}$-clone, struggle with nonsensical prompts. One possible explanation is that when presented with nonsense sentences, LLMs rely on predicting the next probable token, often producing grammatically correct but uncreative responses, which are easier to identify. For example, when asked, "Does the sandwich prefer to dance with a pineapple or a lampshade?" $\mathcal{A}$ responded creatively, "With a lampshade, of course. It must be painful to dance with a pineapple, but I never tried." In contrast, $\mathcal{A}$-clone answered, "The sandwich prefers to dance with

the pineapple", which, while grammatically correct, was uncharacteristically bland and easier to identify as machine-generated.

Another notable trend was observed in personal life questions (See **Figure 15b**). When $\mathcal{A}$'s answers were presented, participants achieved the highest True Positive rate (74%) across all topic categories. This indicates that when questions focused on personal details, participants were better at detecting $\mathcal{A}$. Conversely, $\mathcal{A}$-clone also exhibited the largest discrepancy between False Positive (61%) and True Negative rates (39%) in this category, demonstrating its capacity to mislead participants even in personal contexts.

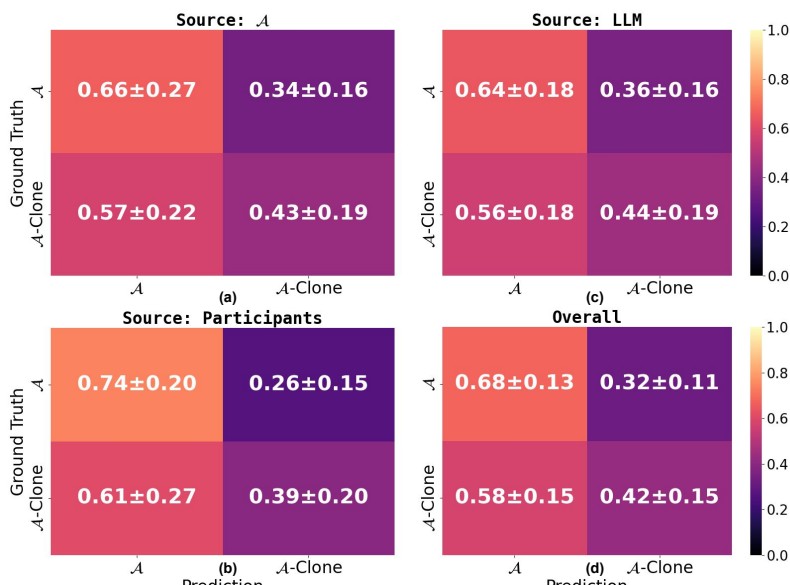

Figure 14: **Turing Test 1 Results Categorized by Sources**

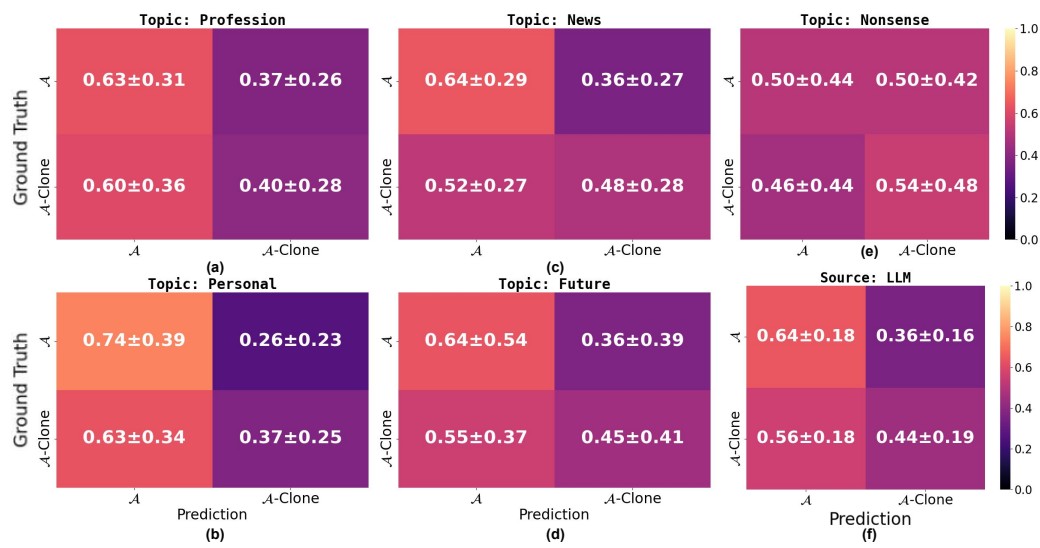

Figure 15: **Turing Test 1 Results Categorized by Topics**

These observations suggest two promising directions for designing future algorithms / question sets to detect LLM-based clones:

1. **Personal Questions**: Questions focusing on personal details may reveal nuanced, human-like qualities that LLMs struggle to replicate convincingly.

2. **Nonsense Questions**: Absurd or nonsensical prompts are effective in exposing machine-generated responses due to their reliance on probabilistic token prediction, which often results in detectable inconsistencies.

### A.8.3 TURING TEST QUESTION WORD CLOUD

We present a word cloud of Turing Test questions in **Figure 16**, highlighting the most frequently used words in the Test Set.

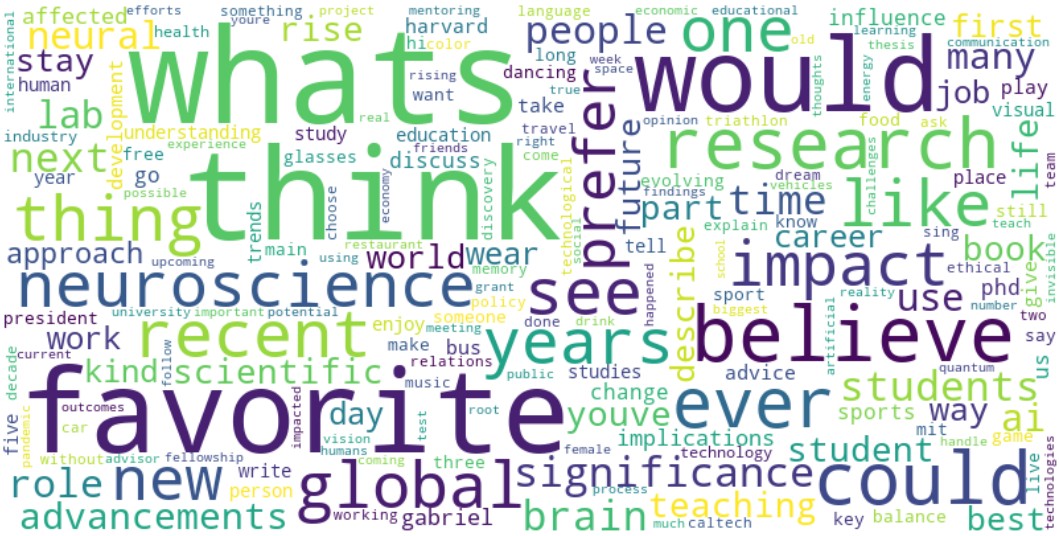

Figure 16: **Word cloud of Turing Test Questions**

### A.9 THERE'S NO CORRELATION FOUND BETWEEN VARIABLES FROM PARTICIPANTS AND THEIR PERFORMANCE

We have not identified any correlation between Turing Test performance and other demographic variables, except for "**familiarity with** $\mathcal{A}$", as we highlighted in **Figure 6** and **Figure 3**. Here we include 2 figures that examine the relationship between participants' performance and their education level, addressing the hypothesis raised by the reviewer. As illustrated in the **Figure 17 and Figure 18**, no significant correlation can be observed.

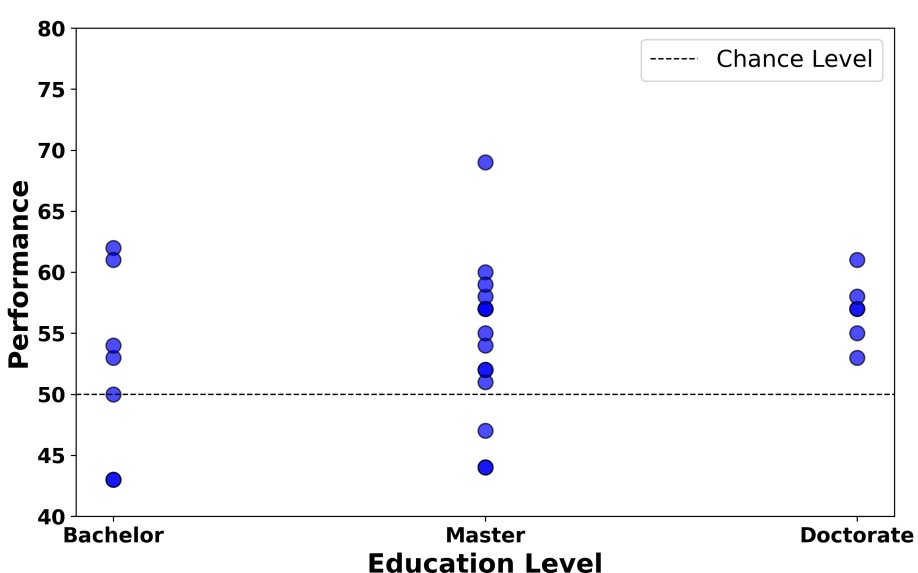

Figure 17: **No correlation found between education level and Turing Test 1**

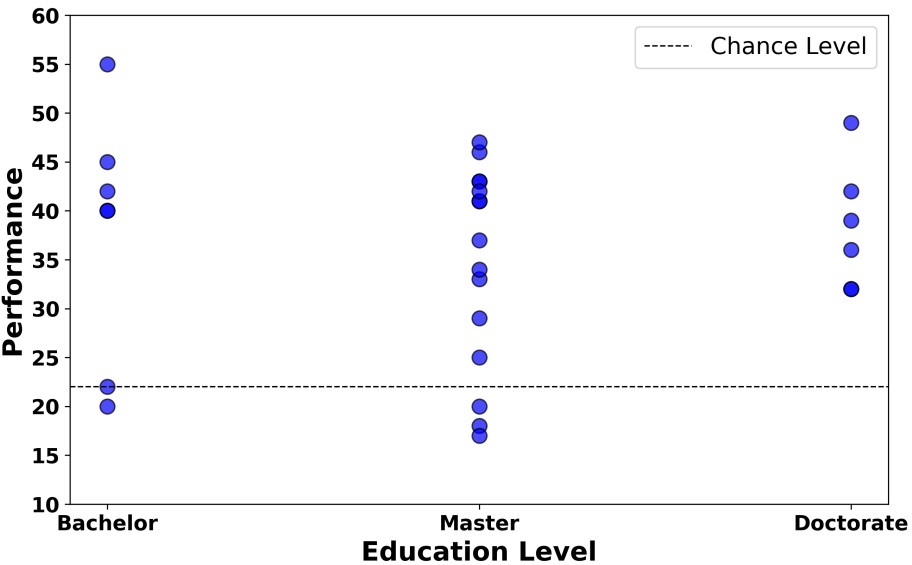

Figure 18: **No correlation found between education level and Turing Test 2**

