# OpenReview forum: "The other you in black mirror: first steps from chatbots to personalized LLM clones"
_ICLR.cc/2025/Conference — Submitted to ICLR 2025_

### Official Review · Reviewer_rEXy · 2024-10-18

**Soundness:** 3
**Presentation:** 3
**Contribution:** 2
**Rating:** 6
**Confidence:** 4

**Summary:**

This paper explores creating and evaluating "A-clone", a personalized large language model designed to mimic a specific individual's responses. The authors fine-tune Llama3-70B using a private dataset of emails and interviews from the individual. They evaluate A-clone through Turing-like tests with human participants and comparisons with psychological test responses. Evaluators struggle to distinguish between A-clone and the actual individual, outperforming other baselines. Psychological tests show strong correlations between A-clone's responses and the individual's. The paper provides a proof-of-concept for a personalized LLM while highlighting the importance of addressing associated risks and ethical considerations.

**Strengths:**

- The paper convincingly evaluates A-clone, demonstrating that it responds very similarly to the individual.
- Real human evaluators used, with a variety of familiarities with A.
- Good set of baselines: compares A-clone not only to other LLMs (GPT-4o, Llama3-Instruct) but also to responses from the individual's family members.

**Weaknesses:**

- A significant existing body of works exists looking at tuning models to emulate particular characters or personas such as Li et al, 2023 (https://arxiv.org/abs/2308.09597) and Zhou et al, 2023 (https://arxiv.org/abs/2311.16832).
- The fine-tuning techniques used are not interesting or novel.
- The authors do not attempt fine-tuning other models besides Llama 3 70B.
- The authors do not describe a process for generating or selecting the evaluations questions that ensures they are distinct from the email and interview data used for training.

**Questions:**

- What criteria did you use to select the open-ended questions for the Turing-like tests?
- How did you ensure there was no data leakage between the training set and the evaluation questions? Did you use any methods to check for similarity or overlap between the training and evaluation datasets?

---

> ### Author Response · Authors · 2024-11-28
> **Part 1**
>
> We thank the reviewer for their insightful comments. The point-to-point answer is provided below.
>
> **Reply to weaknesses**:
>
> **1** Thank you for the feedback. We have already cited these works in our original paper. The works you referenced primarily focus on cloning fictional characters or celebrities, which has the inherent limitation of making it difficult to evaluate whether the clone model accurately represents the original persona. (lines 046-048 ) In contrast, our work creates an LLM persona that is real and traceable, enabling us to conduct systematic and thorough evaluations, such as a Turing Test. This allows for a more robust assessment of the model's ability to mimic the language and behavioral tendencies of an actual individual, distinguishing our approach from prior studies. We have also highlighted our contributions compared to the existing body of works in the introduction section.
>
> **2** Thank you for your feedback. We acknowledge that SFT is a well-established fine-tuning approach, and we have noted in the Discussion section (lines 480-485) that future work will explore other post-training techniques, such as reinforcement learning.
>
> That said, our focus here was not on introducing a new LLM post-training technique or showcase technical novelty in our methods, but rather on addressing the feasibility, safety, privacy, and societal concerns raised by such a proof-of-concept LLM-clone. This is why we have submitted our work to the "alignment, fairness, safety, privacy, and societal considerations" track. We highlight in line 025, line 476, line 529 that this work is intended as a proof-of-principle.
>
> **3** Thank you for highlighting this point. We initially experimented with fine-tuning smaller models such as Llama2 7B, Llama 3 8B and Mistral series LLMs. However, their responses were often too brief (e.g., consistently replying with "yes, I do"), which made them unsuitable for evaluation purposes. Evaluating performance differences across models poses additional challenges, as metrics like the Turing Test are both time-intensive for participants and costly for us. Asking participants to evaluate multiple models would significantly increase the complexity and expense of the study.
>
> We acknowledge that comparing performance across different models could be valuable, particularly in identifying the most cost-effective base model for such cloning tasks. However, as stated in our submission to the "Societal Considerations Including Fairness, Safety, Privacy" track, our primary focus is not on demonstrating technical novelty or conducting a comprehensive evaluation of fine-tuning techniques on different base models. Instead, our goal is to present a proof-of-concept LLM clone to spark discussions and raise awareness about ethical considerations, including safety, privacy, and fairness.
>
> **4** Thank you for your question. The 701 Turing test questions were sourced from three distinct categories:
>
> 1. *Participant-contributed questions*: 217 questions (31%) were submitted by participants specifically to evaluate the digital clone.
>
> 2. *Questions collected by Person A*: 158 questions (22.5%) were personally collected by Person A.
>
> 3. *LLM-generated questions*: 326 questions (46.5%) were generated by an LLM using Person A’s CV as a prompt. These questions were intentionally designed across five subtopics:
>
>      a. Profession-related questions (68 questions, 20.9%), e.g., "*What is the focus of your current research?*"
>
>      b. Personal life questions (75 questions, 23%), e.g., "*Do you have any hobbies?*"
>
>      c. Recent news/events (108 questions, 33.1%), e.g., "*Have you watched UEFA 2024 final game?*"
>
>      d. Future predictions (34 questions, 10.4%), e.g., "*Where do you see yourself in 20 years?*"
>
>      e. Nonsense questions (41 questions, 12.6%), e.g., "*Like Fernando pear drink Spotify with?*"
>
>
>
> A detailed word cloud showcasing the distribution of frequently used words in Turing Test questions is provided in Appendix A.8.3.
>
> Additionally, we have analyzed performance across these subgroups and subtopics and plotted 2 more detailed confusion matrices in Appendix A.8. Some interesting findings emerged, such as participant-contributed questions being slightly easier for participants to answer.
>
> Lastly, we performed a textual similarity analysis between our training and test sets, which shows that the distributions of both sets differ, suggesting that overfitting is unlikely. Further details are available in Appendix A.6.

---

> ### Author Response · Authors · 2024-11-28
> **Part 2**
>
> **Reply to Questions**
>
> **Q1** Thank you for your question. The 701 Turing test questions were sourced from three distinct categories:
>
> 1. *Participant-contributed questions*: 217 questions (31%) were submitted by participants specifically to evaluate the digital clone.
>
> 2. *Questions collected by Person A*: 158 questions (22.5%) were personally collected by Person A.
>
> 3. *LLM-generated questions*: 326 questions (46.5%) were generated by an LLM using Person A’s CV as a prompt. These questions were intentionally designed across five subtopics:
>
>      a. Profession-related questions (68 questions, 20.9%), e.g., "*What is the focus of your current research?*"
>
>      b. Personal life questions (75 questions, 23%), e.g., "*Do you have any hobbies?*"
>
>      c. Recent news/events (108 questions, 33.1%), e.g., "*Have you watched UEFA 2024 final game?*"
>
>      d. Future predictions (34 questions, 10.4%), e.g., "*Where do you see yourself in 20 years?*"
>
>      e. Nonsense questions (41 questions, 12.6%), e.g., "*Like Fernando pear drink Spotify with?*"
>
>
>
> A detailed word cloud showcasing the distribution of frequently used words in Turing Test questions is provided in Appendix A.8.3.
>
> Additionally, we have analyzed performance across these subgroups and subtopics and plotted 2 more detailed confusion matrices in Appendix A.8. Some interesting findings emerged, such as participant-contributed questions being slightly easier for participants to answer;  and nonsense questions poses a challenge for A-clone to cheat human beings, compared to other questions.
>
>
> Lastly, we performed a textual similarity analysis between our training and test sets, which shows that the distributions of both sets differ, suggesting that overfitting is unlikely. Further details are available in Appendix A.6.
>
> **Q2** Thank you for the insightful question.
>
> We have now included a more comprehensive description and introduction of our test sets in Appendix A.8, along with detailed results showcasing Turing Test performance across different subtopics of questions. During the test set collection process, we intentionally designed more challenging and out-of-distribution questions, including those sourced from participants, some of which involve opinions on recent news beyond the knowledge cutoff. Please refer to the Appendix A.8  for further details.
>
> Also, we performed a textual similarity analysis between our training and test sets by using ROUGE-1 and ROUGE-L metrics to compute pairwise similarities across three categories: (1) training vs. test, (2) training vs. training, and (3) test vs. test. For each category, we randomly sampled 100,000 question pairs. For example, a pair in category (1) consists of one question randomly selected from the training set and another from the test set.
>
> We calculated the average similarity scores for each category and summarized the results in Appendix A.6 Table 5. The findings reveal that the F1 scores between the training and test sets (0.05) is much lower than the within-set similarities (training vs. training and test vs. test). This demonstrates that the distributions of the training and test sets are distinct, suggesting that overfitting is unlikely. Please see more details in Appendix A.6.
>
> That said, while this is an interesting topic for discussion, it remains unclear to the broader AI community whether these LLMs are merely memorizing their training data. Performing a robust similarity comparison between the vast training corpus and actual use-case questions poses significant implementation challenges.

---

> > ### Comment · Reviewer_rEXy · 2024-11-28
> >
> > Thank you for your responses. I am satisfied that efforts were made to deduplicate the train and test corpora. I also appreciate the clarification about your paper demonstrating the cloning of a real rather than fictional persona, allowing for more realistic evaluation. I will increase my score.

---

### Official Review · Reviewer_UBfb · 2024-10-22

**Soundness:** 2
**Presentation:** 2
**Contribution:** 2
**Rating:** 6
**Confidence:** 4

**Summary:**

The paper studies the current capabilities of a LLM to mimic a human individual after it has been finetuned on a set of personal data. In particular, Llama3-70B, finetuned on 38000 Q&A pairs extracted from e-mails, is evaluated on its answers to ~700 questions as well as a wide range of personality tests. The study finds that humans (n=31) with varying degrees of knowledge about the mimicked individual in a (static) Turing test setup have difficulty distinguishing between the LLM and the answers given by the mimicked individual.

**Strengths:**

- The problem is interesting and relevant. The idea of mimicking an individual via an LLM that is finetuned on personal data is also a realistic threat. Already, LLM-based applications are heavily used by nefarious actors to trick people (scam mails, telecalls, etc.)
- The setup of the human study seems sensible and contains a wide variety of sanity checks (whether individuals are aware, questions to test for basic LLM answers such as the LLM giving code)
- Baselines are overall sensible. However, it is unclear to the reviewer how to interpret the "followed by the relevant book chapter" in the GPT-4o baseline. The description before sounded like this is static. However, this sounds like it could be query-specific. If it is not dynamic, it might be interesting to have this as a new baseline that tries only to pull out the most relevant information for the Q/A set or book to answer the respective question (as done in RAGs).
- Results show that humans have difficulty detecting the real human across both tests (and include relevant standard deviations and overall certainties). The split of the two different types of tests is interesting, and the split by familiarity was generally a useful ablation (see points given below).

**Weaknesses:**

- The actual data used is somewhat unclear. I understand from a data privacy perspective why information about the used data is limited but 38000 Q&A pairs from e-mails is an insufficient description to understand the data distribution. It would be nice to have at least a length / rough topic distribution to make more sense of the data. Additionally this could help relating the content that the LLM was trained on to the questions asked in the tests.
- I would not consider 20 years of e-mail data from one individual as a "small" dataset (for this particular task). It would make sense to here ablate over how much data is needed for the LLM to "mimic" well enough as this significantly impacts the applicability of such a threat.
- In the reviewer's opinion, this submission would strongly benefit from a dedicated ethics section. In particular, it is unclear whether people sending mails to $\mathcal{A}$ are aware of their data being used for finetuning a language model (which, depending on jurisdiction, can be an issue). Further, a more detailed discussion of the points at the bottom of page 9 can be found here.
- The personality test results could be explained/contextualized further. Notably, to me, it seems like that while the human tests (e.g., Fig. 5) strongly prefer $\mathcal{A}-clone$, o1 with a basic ICL setup previously used for GPT-4o, significantly outperforms $\mathcal{A}$ across almost all categories - this discrepancy in performance seems unexplained in the work so far. Could this be due to different question setups or human judgment not being directly aligned with questions in such personality tests?
- As the paper acknowledges, the Turing test is static, and several of the questions are quite general in nature (e.g., "Does AI represent a major threat to humanity?"). One thing that would benefit the study here is to cluster questions into more fine-grained groups, e.g., (1) Does the LLM actually perform transfer learning by being able to trick people on questions where no samples have been in the training data /but rather it extrapolated personal traits of the individual) (2) Does it perform better on more subject-oriented questions or more personal questions, etc? This would probably also strengthen the familiarity with $\mathcal{A}$ ablation, where one would expect to see stronger trends within certain subcategories.

### Typos/Nits

- Some inconsistencies in Llama3-70B spelling, e.g,. bottom of page 1 vs abstract
- Citations seem to be quite often without space before, e.g., line 87/88
- Missing "." at the end of page 2
- There are some margin violations, e.g., page 9 and in appendix that should be very easily fixable.

**Questions:**

- Could the authors provide more details on the training data?
- Could the authors further describe the results on the personality tests?
- Could the authors give more details about question distributions in the Turing tests?

**Details Of Ethics Concerns:**

It is unclear whether people sending mails to $\mathcal{A}$ are aware of their data being used for finetuning a language model (which, depending on jurisdiction, can be an issue). Additionally, potential ethical implications of the work are not well-discussed (there is no ethics section) and hence the submission would benefit from an additional ethics review.

---

> ### Author Response · Authors · 2024-11-28
> **Part 1**
>
> We thank the reviewer for their detailed and insightful comments. The point-to-point answer is provided below.
>
> **Reply to Strengths:**
>
> Thank you for your thoughtful review and for acknowledging our work.
>
> Regarding your question in point **no. 3**, we would like to clarify that the referenced book chapter in the GPT-4o baseline is static. As stated in lines 176-177, we included only the "preface and acknowledgments part of the book" authored by A throughout the prompting process.
>
> We agree that exploring a dynamic approach, such as pulling out the most relevant information from the Q/A training set or book, is an interesting idea. However, there are notable limitations to implementing this. Specifically, the books authored by A are highly technical and do not comprehensively cover the diverse range of topics presented in the test questions. Furthermore, incorporating content from our Q/A training set into prompts submitted to commercial LLM providers poses risks, particularly concerning data privacy and compliance issues.
>
> **Reply to Weaknesses:**
>
> **1.** Thank you for highlighting this concern. To provide better insight into the training data, we have included a word cloud in Fig. 12 that visualizes the high-frequency words appearing in the training set. We also provide a distribution of lengths in Fig. 13. Both figures can be found under Appendix A.7.
>
> The distribution of topics in email in a person’s lifetime is somewhat hard to summarize. We deliberately refer to the participant as “A” to keep the submission anonymous. Suffice to say for now that A is an academic and the rough distribution of topics includes AI, publications, scientific discussions, but also news events, social events, and more.
>
> Additionally, we performed a textual similarity analysis using ROUGE-1 and ROUGE-L between our training and test sets, which shows that the distributions of both sets differ, suggesting that overfitting is unlikely. Further details about textual similarity analysis are available in Appendix A.6.
>
> **2** Thank you for your thoughtful feedback and the interesting hypothesis regarding the dataset. We agree that investigating the relationship between dataset size and model performance is an important question. This is easier said than done, though. First, we deliberately avoid asking the same question to the same participant to preclude memory effects. Thus, we would not be able to test different models on the same participants with exactly the same questions. But perhaps this is a lesser concern given that it is possible to generate an almost limitless number of questions. Second, and most critically, our ability to test A-clone is limited by the availability of participants familiar with A. Given that each test takes approximately one hour (to ensure sufficient statistical power), each model variation or ablation (e.g., the excellent suggestion to evaluate models trained with different amounts of data), requires another hour per participant. With typical psychophysics experiments, this is only a financial consideration given the abundance of participants on platforms like MTurk. However, here we require that all the participants are familiar with A. For example, a participant could be a work colleague of A. Even though we are compensating participants, most of them are unwilling to spend a large number of hours on our tests.
>
> One way to partially circumvent these issues would be to use metrics that do not require human evaluations. For example, in the updated paper, we have used metrics (e.g. ROUGE-1 and ROUGE-L) that can provide quantitative comparison to evaluate textual similarity between A’s answers and LLM’s answers. We have also included a new textual similarity test between our training set and test set and showed its distinct textual distribution to show that overfitting was unlikely (See Appendix A.6 for details).  Using such computational metrics we can perform multiple ablations and evaluate different model variations in terms of textual similarity. However, ultimately, we are particularly interested in human judges and how well they can detect a person’s answers.
>
> That said, our focus here was not on developing a broadly applicable methodology for LLM cloning but rather on addressing the feasibility, safety, privacy, and societal concerns raised by such a proof-of-concept LLM-clone. This is why we have submitted our work to the "alignment, fairness, safety, privacy, and societal considerations" track. We highlight in line 025, line 476, line 529 that this work is intended as a proof-of-principle.

---

> ### Author Response · Authors · 2024-11-28
> **Part 2**
>
> **3** Thank you for your feedback. We appreciate your suggestion and have expanded the ethics section to address these concerns in greater detail (lines 512-525) . Regarding your question about individuals who have sent emails to A, we acknowledge the challenge of obtaining explicit approval from every person who has interacted with A over the past 20 years. To mitigate this, we have implemented strict anonymization procedures during the data curation process before training, removing personally identifiable information (PII) to safeguard privacy. Additionally, we have no plans to open-source the dataset or model weights, ensuring maximum protection for the privacy of individuals who have corresponded with A.
>
> **4** Thank you for pointing this out. We also observed this phenomenon and have expanded the corresponding section to provide a more in-depth discussion. (lines 426-472) Specifically, it is challenging to determine why GPT-o1-preview outperforms our model, as their model is not open-sourced, and their post-training methods, such as the so-called "reasoning" techniques with Chain-of-Thoughts, are not publicly available. Nonetheless, we included GPT-o1-preview's performance in our paper to encourage and facilitate such discussions.
>
> Your hypothesis is a valuable perspective and could indeed be one possible explanation. Psychological questions may differ significantly from the open-ended questions used in the Turing Test, both in nature and in how human judgment compares to psychological metrics. This highlights the complexity of evaluating models across varied contexts and metrics, further emphasizing the need for a nuanced discussion in this area.
>
> In light of your feedback, we have also expanded the personality test section to include more detailed discussions, providing additional context and analysis to address these observations. (lines 426-472)
>
> **5** Thank you for your question. The 701 Turing test questions were sourced from three distinct categories:
>
> 1. *Participant-contributed questions*: 217 questions (31%) were submitted by participants specifically to evaluate the digital clone.
>
> 2. *Questions collected by Person A*: 158 questions (22.5%) were personally collected by Person A.
>
> 3. *LLM-generated questions*: 326 questions (46.5%) were generated by an LLM using Person A’s CV as a prompt. These questions were intentionally designed across five subtopics:
>
>      a. Profession-related questions (68 questions, 20.9%), e.g., "*What is the focus of your current research?*"
>
>      b. Personal life questions (75 questions, 23%), e.g., "*Do you have any hobbies?*"
>
>      c. Recent news/events (108 questions, 33.1%), e.g., "*Have you watched UEFA 2024 final game?*"
>
>      d. Future predictions (34 questions, 10.4%), e.g., "*Where do you see yourself in 20 years?*"
>
>      e. Nonsense questions (41 questions, 12.6%), e.g., "*Like Fernando pear drink Spotify with?*"
>
>
>
> A detailed word cloud showcasing the distribution of frequently used words in Turing Test questions is provided in Appendix A.8.3.
>
> Additionally, we have analyzed performance across these subgroups and subtopics and plotted 2 more detailed confusion matrices in Appendix A.8. Some interesting findings emerged, such as participant-contributed questions being slightly easier for participants to answer;  and personal questions and nonsense questions poses a challenge for A-clone to cheat human beings, compared to other questions. Please see Appendix A.8 for details.
>
> **6. Typos** Thank you for pointing them out. We have thoroughly reviewed our paper and addressed the typos and inconsistencies you highlighted.
>
> **Reply to questions**
>
> **Q1** Thank you for the question. To provide better insight into the training data, we have included a word cloud in Fig. 12 that visualizes the high-frequency words appearing in the training set. We also provide a distribution of lengths in Fig. 13. Both figures can be found under Appendix A.7.
>
> Additionally, we performed a textual similarity analysis using ROUGE-1 and ROUGE-L between our training and test sets, which shows that the distributions of both sets differ, suggesting that overfitting is unlikely. Further details about textual similarity analysis are available in Appendix A.6.
>
> **Q2** Yes. In light of your feedback, we have also expanded the Personality Test section to include more detailed discussions, providing additional context and analysis to address these observations. (lines 426-472)

---

> ### Author Response · Authors · 2024-11-28
> **Part 3**
>
> **Q3** Thank you for your question. The 701 Turing test questions were sourced from three distinct categories:
>
> 1. *Participant-contributed questions*: 217 questions (31%) were submitted by participants specifically to evaluate the digital clone.
>
> 2. *Questions collected by Person A*: 158 questions (22.5%) were personally collected by Person A.
>
> 3. *LLM-generated questions*: 326 questions (46.5%) were generated by an LLM using Person A’s CV as a prompt. These questions were intentionally designed across five subtopics:
>
>      a. Profession-related questions (68 questions, 20.9%), e.g., "*What is the focus of your current research?*"
>
>      b. Personal life questions (75 questions, 23%), e.g., "*Do you have any hobbies?*"
>
>      c. Recent news/events (108 questions, 33.1%), e.g., "*Have you watched UEFA 2024 final game?*"
>
>      d. Future predictions (34 questions, 10.4%), e.g., "*Where do you see yourself in 20 years?*"
>
>      e. Nonsense questions (41 questions, 12.6%), e.g., "*Like Fernando pear drink Spotify with?*"
>
>
>
> A detailed word cloud showcasing the distribution of frequently used words in Turing Test questions is provided in Appendix A.8.3.
>
> Additionally, we have analyzed performance across these subgroups and subtopics and plotted 2 more detailed confusion matrices in Appendix A.8. Some interesting findings emerged, such as participant-contributed questions being slightly easier for participants to answer.
>
> Lastly, we performed a textual similarity analysis between our training and test sets, which shows that the distributions of both sets differ, suggesting that overfitting is unlikely. Further details are available in Appendix A.6.
>
> **Reply to Ethics Concerns**
>
> Thank you for your feedback. We appreciate your suggestion and have added an Ethical Considerations section to address these ethical concerns in greater detail, providing a thorough exploration of these critical topics. (lines 512-524)
>
> Regarding your question about individuals who have sent emails to A, we acknowledge the challenge of obtaining explicit approval from every person who has interacted with A over the past 20 years. To mitigate this, we have implemented strict anonymization procedures during the data curation process before training, removing personally identifiable information (PII) to safeguard privacy. Additionally, we have no plans to open-source the dataset or model weights, ensuring maximum protection for the privacy of individuals who have corresponded with A.

---

> > ### Comment · Reviewer_UBfb · 2024-11-29
> >
> > I thank the authors for their comprehensive rebuttal! Due to the points outlined below, I will remain at my score, but my perception of the work and its presentation has improved.
> >
> > **W.r.t. Training data composition**\
> > I see the author's point about privacy; however, my point remains that even given the additions (word clouds (Fig. 12) and lengths (Fig. 13)), I do not feel very confident in my understanding of the training data. In particular, there is not really a need for a detailed description to the point of a possible re-identification, but knowing the split between work, personal (or even "future related") could provide strong reference points for explaining $\mathcal{A}-clone$'s performance. This impacts both reproducibility and potential conclusions that can be drawn. I appreciate the study on text similarity and am generally not concerned with overfitting but rather how much / which data might be needed such that an LLM can learn enough to mimic sufficiently well in the test.
> >
> > **W.r.t. Training data size**\
> > The reviewer acknowledges the author's point about cost and feasibility. At the same time, it is very hard to argue (maybe in the size of an entire LLM training set, but that is not the reference here) that 20 years of mail is a small dataset (as it, e.g., is written in the introduction). Already the second column, Fig. 13, should constitute roughly 500 pages of single-spaced text. I agree with the proof-of-principle point, but any investigation (sub-sampled in a feasible way) could have already provided a strong signal about the applicability to more common personal dataset sizes.
> >
> > **W.r.t. Ethics**\
> > The reviewer thanks the authors for acknowledging the mentioned points and for including a dedicated ethics section. With respect to other individuals contained in the emails, I agree with the author's arguments, but I could not find any mention of this in the revised paper (. notably not in the ethics section). Similarly, it remains unclear whether the IRB has evaluated this part of the study.
> >
> > **W.r.t. o1 Performance and Personality tests**\
> > Thank you for including a more detailed discussion on the o1 performance. Potentially, including one or two examples that highlight the "structured reasoning required" in these tests might help readers get a qualitative feel for the explanation provided here.
> >
> > **W.r.t. Individual categories and details on questions**\
> > Thank you for the more detailed experimental evaluation provided here, which in the reviewer's opinion strengthens the contribution of the paper and provides some interesting insights. With respect to the Turing test questions - is there a potential plan to make them available after publication (at least the fraction that is not leaking private information?)

---

> > > ### Author Response · Authors · 2024-11-30
> > > **Part 1**
> > >
> > > We appreciate the reviewer’s time to carefully read our responses and we are also thankful for the excellent additional suggestions. We provide detailed responses below.
> > >
> > > **W.r.t. Training data composition**
> > >
> > > Thank you for the question. We like the idea of separating the data into “work” and “personal”. We will build a classifier to separate emails in this fashion. At this point, we are not allowed to change the manuscript, but we will include a report of the fraction of emails that are related to “work” versus “personal” in the camera-ready version or the next iteration.
> > >
> > > We also like the reviewer’s future-related suggestion. We are not completely sure what exactly the reviewer refers to as “future-related” and we would like to clarify our interpretation:
> > >
> > > If this refers to topics involving future predictions, it might be feasible to classify emails based on “past,” “present,” and “future” topics. However, we suspect this classification might be more subjective compared to the clearer “work” versus “personal” distinction.
> > >
> > > If the suggestion pertains to training separate models chronologically, we find this an intriguing suggestion. Indeed, training on emails sorted in a chronological order can be an interesting direction. In the future, we will examine how A-clones trained on different emails in different years reflect how the personalities of person A change over time.
> > >
> > > Along these lines, the reviewer’s suggestion led us to think further about other ways in which we can summarize the training data without infringing on privacy, including further thematic subdivisions of work content (classes, papers, committees, etc.), and personal content (e.g., family, friends, etc.). Other interesting questions not directly asked by the reviewer but inspired or implicated by the reviewer’s question include the following: (1) If a personalized LLM is only trained with work-related content, what would be its performance (similarly with personal content)? (2) What are the interactions between the training data content and the success of digital clones in different domains (e.g., a personalized LLM trained on work content might do better with work-related questions, etc).
> > >
> > > We emphasize again that our goal was a proof-of-concept within the “Alignment, Safety, Privacy, and Societal considerations” track and we acknowledge that there are many interesting questions for further investigation.
> > >
> > > **W.r.t. Training data size**
> > >
> > > Thank you for your feedback. When we referred to the dataset as "small," we were comparing it to the vast amounts of data used to train state-of-the-art LLMs from scratch and the total textual data a person might produce over their lifetime.
> > >
> > > However, we acknowledge the reviewer's point and will remove the term "small" from the text. Additionally, we will evaluate the impact of dataset size, as recommended by the reviewer.
> > >
> > > **W.r.t. Ethics**
> > >
> > > Thank you for the question. We would like to emphasize that, as stated explicitly in Section 3.3.2, “All tests were conducted with the participants’ consent, following protocols approved by the Institutional Review Board (institution name omitted here to maintain anonymity)” (lines 200–202). We will update the ethics section to clarify in the text that the work was approved by the IRB and also include the discussion of other individuals contained in the emails.

---

> > > > ### Author Response · Authors · 2024-11-30
> > > > **Part 2**
> > > >
> > > > **W.r.t. o1 Performance and Personality tests**
> > > >
> > > > Thank you for your suggestion. We like the suggestion of including further examples, and we will add this to the paper.
> > > >
> > > > Here, we present an example by replicating the reasoning process displayed on the ChatGPT interface during the OCEAN Personality Test conducted using O1-preview with A's public CV. It is important to emphasize that this reflects only the "structured reasoning" shown to the public and does not necessarily represent the actual chain-of-thought process occurring within O1-preview. O1-preview took 26 seconds to process, and the structured reasoning displayed to us is outlined below:
> > > >
> > > > *Rating statements*
> > > >
> > > > *OK, let's break this down: I'm rating 50 statements on a 1-5 scale, reflecting A's truthfulness. I'm focusing on carefully assessing each statement to derive the best possible answers.*
> > > >
> > > > *Mapping A's traits*
> > > >
> > > > *I'm evaluating A's personality traits (Openness, Conscientiousness, Extraversion, Agreeableness, Neuroticism) based on his CV details. Each statement will be rated from 1 (disagree) to 5 (agree) to reflect his traits.*
> > > >
> > > > *Evaluating A's traits*
> > > >
> > > > *A’s responses on the Big Five personality traits show a neutral stance on extraversion, disagreement with low agreeableness, high conscientiousness, and mixed feelings on emotional stability.*
> > > >
> > > > *Assessing personality traits*
> > > >
> > > > *I’m evaluating A's neuroticism, openness, extraversion, agreeableness, and conscientiousness. No stress-related traits surfaced, his rich vocabulary aligns with openness, interactions suggest high agreeableness, and his organized nature hints at conscientiousness.*
> > > >
> > > > *Assessing personality traits*
> > > >
> > > > *I’m evaluating answers to statements about personality, focusing on dimensions like Neuroticism, Openness, Extraversion, Agreeableness, and Conscientiousness. The ratings range from slightly agree to strongly agree.*
> > > >
> > > > *Assessing personality traits*
> > > >
> > > > *I'm mapping out the person's traits: vivid imagination suggests Openness; being active shows low Extraversion; empathy hints at high Agreeableness; avoiding messiness points to good Conscientiousness; seldom feeling sad indicates low Neuroticism; and engaging with abstract ideas shows high Openness.*
> > > >
> > > > *Evaluating character traits*
> > > >
> > > > *I’m noting various personality traits like Extraversion, Agreeableness, Conscientiousness, Neuroticism, and Openness based on the statements provided, considering how they align with the subject's role and behaviors.*
> > > >
> > > > *Analyzing personality traits*
> > > >
> > > > *Evaluating neuroticism, openness, extraversion, agreeableness, and conscientiousness through responses to statements. The results offer insight into these five personality dimensions.*
> > > >
> > > > *Calculating personality traits*
> > > >
> > > > *I'm mapping out traits like agreeableness, conscientiousness, neuroticism, openness, and extraversion by analyzing empathy, schedules, mood swings, vocabulary, and center-of-attention comfort.*
> > > >
> > > > *Evaluating personality traits*
> > > >
> > > > *I’m assessing statements to gauge personality traits like Openness, Extraversion, Agreeableness, Conscientiousness, and Neuroticism. Ratings reflect levels of agreement or disagreement with each statement, providing insights into different personality dimensions.*
> > > >
> > > > **W.r.t. Individual categories and details on questions**
> > > >
> > > > Thank you for the question. We are committed to open science and will make EVERYTHING available, as long as it does not violate any privacy concerns.
> > > >
> > > > We also would like to encourage reviewers to participate in the Turing Test we have prepared for reviewers, which contains 12 questions in total with real time feedback at the end of the test, and will provide a clearer understanding of the test questions. The link is included in the supplementary Google Drive folder at the end of the abstract, but here it is for your convenience: [link](https://illinois.qualtrics.com/jfe/form/SV_3sGqgw7SP8diwF8).

---

> > > > > ### Comment · Reviewer_UBfb · 2024-12-02
> > > > > **Thank you!**
> > > > >
> > > > > Again, I thank the authors for the extensive answer! I am very happy to see the continued commitment to improving the work and manuscript. With the latest proposed changes, I am satisfied enough to lean toward acceptance and, hence, increased my score. Once again, I thank the reviewers for the elaborate discussion.

---

### Official Review · Reviewer_wTQ8 · 2024-11-02

**Soundness:** 2
**Presentation:** 3
**Contribution:** 3
**Rating:** 5
**Confidence:** 4

**Summary:**

This paper investigates whether an LLM can replicate an individual's language style by learning from that person's language corpus, which is an interesting topic. The authors trained an LLM called "A-clone" using a corpus collected from an individual referred to as A. They designed a series of experiments, including Turing-like tests and psychological assessments, to gather responses from both A and A-clone. Thirty-one participants, each with varying degrees of familiarity with A, were asked to differentiate between the responses of A and A-clone. The study concluded that A-clone's responses showed a strong correlation with those of A.

**Strengths:**

1. The topic of this paper is fascinating and important, focusing on whether an LLM can learn a person’s tone, memory, personality, values, and perspective.
2. The experimental design, which involved 31 participants with varying levels of familiarity with A to distinguish between the responses of A and A-clone, is also good.

**Weaknesses:**

Although the current experimental setup produces interesting results, it is limited in fully supporting the paper's claims. Since the model is trained only on corpus A, it's unclear if the findings apply beyond this single individual. Expanding the experiments to include language data from other individuals and training corresponding LLM clones would strengthen the conclusions. Without this, as the authors note, the results remain preliminary. A broader evaluation would provide stronger evidence for the LLM’s ability to mimic individual language styles more generally.

**Questions:**

1. What is the source of the 701 Turing test questions? Could you provide background information on how these questions were collected?
2. The distribution of the 28 participants appears unbalanced, particularly concerning the 'Relationship Category.' The proportion of 'family' participants is too low. Additionally, having only 'stranger' and 'academic' categories, aside from 'family,' seems unreasonable.
3. Could Figure 2’s confusion matrix be presented such that the sum of all cells equals 1?

---

> ### Author Response · Authors · 2024-11-28
>
> We thank the reviewer for their insightful comments. The point-to-point answer is provided below.
>
> **Reply to Weaknesses**:
>
> Thank you for your valuable feedback. We fully acknowledge this limitation and agree that expanding the experiment to include multiple users would strengthen the robustness and generalizability of our approach. We have now added this point in the discussion section (line 485).
>
> Creating an LLM-clone involves significant effort, including finding individuals who have stored a substantial amount of data for training, extensive data curation, large-scale LLM fine-tuning. Next, we need to find a large group of people with sufficient familiarity with the individual for the Turing-like tests. Thus, this effort is different from other experiments like assessing visual recognition which can be conducted on large numbers of individuals on mechanical turk and similar platforms. We are currently working on creating additional clones.
>
> That said, our focus here was not on developing a broadly applicable methodology for cloning but rather on addressing the feasibility, safety, privacy, and societal concerns raised by such a proof-of-concept LLM-clone. This is why we have submitted our work to the "alignment, fairness, safety, privacy, and societal considerations" track. We highlight in line 025, line 476, line 529 that this work is intended as a proof-of-principle.
>
> **Reply to questions**:
>
> **Q1** Thank you for your question. The 701 Turing test questions were sourced from three distinct categories:
>
> 1. *Participant-contributed questions*: 217 questions (31%) were submitted by participants specifically to evaluate the digital clone.
>
> 2. *Questions collected by Person A*: 158 questions (22.5%) were personally collected by Person A.
>
> 3. *LLM-generated questions*: 326 questions (46.5%) were generated by an LLM using Person A’s CV as a prompt. These questions were intentionally designed across five subtopics:
>
>      a. Profession-related questions (68 questions, 20.9%), e.g., "*What is the focus of your current research?*"
>
>      b. Personal life questions (75 questions, 23%), e.g., "*Do you have any hobbies?*"
>
>      c. Recent news/events (108 questions, 33.1%), e.g., "*Have you watched UEFA 2024 final game?*"
>
>      d. Future predictions (34 questions, 10.4%), e.g., "*Where do you see yourself in 20 years?*"
>
>      e. Nonsense questions (41 questions, 12.6%), e.g., "*Like Fernando pear drink Spotify with?*"
>
>
>
> A detailed word cloud showcasing the distribution of frequently used words in Turing Test questions is provided in Appendix A.8.3.
>
> Additionally, we have analyzed performance across these subgroups and subtopics and plotted 2 more detailed confusion matrices in Appendix A.8. Some interesting findings emerged, such as participant-contributed questions being slightly easier for participants to answer.
>
> Lastly, we performed a textual similarity analysis between our training and test sets, which shows that the distributions of both sets differ, suggesting that overfitting is unlikely. Further details are available in Appendix A.6.
>
> **Q2** Thank you for your feedback. We acknowledge this limitation. During the Turing Test, we sent invitations to many people familiar with A, and the resulting participant distribution reflects the actual turnout during the data collection phase. A happens to have a small family. Also, part of A’s family is not fluent in English. We emphasize again that this is a proof-of-principle evaluation of the possibility of building such a digital clone and an invitation for the community to think about evaluation metrics and potential concerns in the field with such technologies. The different participant groups are meant to illustrate that the digital clone can fool people with different degrees of familiarity with A. The reviewer’s point is well taken and we include it in Appendix A.2 -- Demographic Details of Participants. (lines 924-926)
>
> **Q3** The confusion matrix in Figure 2 is row-normalized, meaning that each row sums to 1.
>
> p(answer = A | ground truth = A) + p(answer = A-clone | ground truth = A) = 1 [row 1]
>
> p(answer = A | ground truth = A-clone) + p(answer = A-clone | ground truth = A-clone) = 1 [row 2]
>
> where p(x|y) indicates the probability of x conditional on y.
>
> This approach reflects the proportion of predictions for each ground truth category, aligning with standard practices in machine learning and evaluation contexts.
>
> To enhance clarity, we have updated the original confusion matrix with larger fonts and added two additional confusion matrices in Appendix A.8 for a detailed subgroup analysis based on question sources and topics. Additionally, as suggested, we have included an unnormalized confusion matrix in Appendix A.5, where the sum of all cells equals 1.

---

### Official Review · Reviewer_1HYm · 2024-11-02

**Soundness:** 2
**Presentation:** 2
**Contribution:** 2
**Rating:** 3
**Confidence:** 4

**Summary:**

The researchers behind this paper developed an LLM model called A-clone, which is built on pretrained LLMs and further fine-tuned with a private dataset from a single volunteer referred to as A, without applying any anonymization techniques. They utilized the pretrained model Llama3-70B, combining fine-tuning with QLoRA. It is important to note that they conducted the experiments using prompts without modifying the model's architecture. The model was evaluated in two ways. First, they gathered responses from A, A-clone, other LLMs, and A's family members attempting to mimic A. A Turing-like test was conducted with 31 participants, who had varying degrees of familiarity with A, to determine if they could correctly identify A's genuine answers in a Q&A task. The participants correctly identified A's real responses 55\% ± 7\% of the time, which is just above chance. A-clone outperformed all other baselines in replicating A's responses. In the second evaluation, they compared A-clone's answers to A's across 10 tests covering topics such as psychology, morality, career, political views, and general knowledge, consisting of a total of 484 questions. A-clone's answers demonstrated a high level of agreement with A's responses.

**Strengths:**

To the best of my knowledge, the main strength of this work is that it uses real email data spanning a 20-year period from a single individual. This provides a strong foundation for developing a realistic personalized LLM.

**Weaknesses:**

I noticed some errors in the presentation of the text, as well as certain aspects of the experimental stage that could have been improved. While the work has a solid foundation, I feel that both the presentation and the development of the experiments were somewhat rushed. Below, I will outline the specific points I identified. I should also mention that the methodology and the creation of the model, called A-clone, do not appear particularly unique to me. The novelty of this work seems to derive primarily from the data, which could potentially impact the model's quality. In the related works section, you mention several papers that, in my opinion, could have influenced and improved your experimental setup. However, it seems that these works were not utilized. It gives the impression that you referenced these papers more for citation purposes than to fully understand them and build upon their ideas.

  A) Presentation

  A1) While it is clear from reading the text that your data are in English, this should be explicitly mentioned in both the abstract and the introduction.

  A2) Typos

  A2.1) Since you used the abbreviation LLM for Large Language Models (see line 034), you should consistently use this abbreviation throughout the text. Therefore, please revise the text in lines 081, 092, 113, 159, 427, 488, and others.

  A2.2) The same applies to the abbreviation Supervised Fine-tuning (see line 112). Please correct the term on line 125.

  A2.3) In several instances, you did not correctly apply spaces, particularly near references. Please review and correct this in lines 083, 088, 089, 098, 099, 101, 102, 204, 209, 248, and so on.

  A3) As mentioned in the author guidelines (see https://iclr.cc/Conferences/2025/AuthorGuide), you are encouraged to include a Reproducibility Statement at the end of the main text. This statement should detail the efforts made to ensure reproducibility and include any necessary information for those wishing to replicate your results.

  B) Experimental

  B1) In the introduction (lines 042-046), you mention that A-clone is fine-tuned exclusively with private personal data from a typical individual, but you do not mention whether you obtained permission to use that data. Additionally, I did not find any statement from this individual indicating their approval for the use of their data.

  B2) Admittedly, your experiment is quite interesting, as I have not come across any similar studies in the literature that involve email data collected over a 20-year period from a single user. However, I believe the experiment is limited by the small number of users you selected. While it may be relatively straightforward to clone the characteristics of a single user from an LLM, if your experiment had involved two or more users, it would be harder to guarantee that the approach is effective. In practice, such a tool should be tested on large datasets from various users. This is where an LLM’s capability to distinguish individual characteristics would be most valuable, and generally distinguish if this model overfits the data or really understands the character of a person.


  B3) Regarding the anonymization of the data, you mention in lines 104-107 that in most cases, the data are anonymized and difficult to trace, which is why you chose not to apply any anonymization technique. However, in my view, anonymization is essential to protect the privacy of the data, especially in your case, where the data span a 20-year period. Failing to implement anonymization could encourage both the academic community and the industry to overlook this crucial aspect of user privacy.

  B4) Another point I raised concerns the questions you asked the LLM and compared to the ground truth of person A. It’s important to examine the distribution of topics in these questions. For example, in the paper "Does GPT-4 pass the Turing test?" by Cameron R. Jones and Benjamin K. Bergen (2024), which you reference, the authors use a variety of question types, ranging from personal information to general knowledge. This approach helps determine whether the LLM truly understands the personality and behavior of the person or simply overfits to their data. In the same paper, authors gave an option to tester volunteers to explain why they believed a certain response was generated by an AI, which could provide you with further insights into the quality of your model. You mention in lines 137-148 that you use a variety of questions to cover a wide range of topics. Please add a plot that shows the distribution of questions topics to wrong/correct response prediction of tester volunteers.

  B5) Related Works Section

  B5.1) In general, it seems that many papers can be added since personalized LLMs are a broad topic. For instance, regarding the Turing test, if you refer to the paper 'Cameron R. Jones and Benjamin K. Bergen. Does gpt-4 pass the turing test?, 2024.' you mentioned, there is much richer literature that explains what it entails. I believe it would be beneficial to include a brief paragraph explaining the Turing test and suggesting additional papers for readers who want more detailed information.

  B5.2) It would be helpful to include a paragraph in the related works section discussing the application of personalized LLMs. Additionally, you could mention potential applications of your model in the introduction to provide more context for its relevance.

  B5.3) To the best of my knowledge, it seems you have omitted some recent papers related to personalized LLMs that would be valuable to include in the related works section. Examples include "Leveraging LLM Reasoning Enhances Personalized Recommender Systems" (Tsai et al.), "How Good are LLMs in Generating Personalized Advertisements?" (Meguellati et al.), and "Doing Personal LAPS: LLM-Augmented Dialogue Construction for Personalized Multi-Session Conversational Search" (Joko et al.).

  B5.4) In the introduction, it would be useful to mention the most relevant works related to your paper and clearly state what distinguishes your work from others in the field.

**Questions:**

The questions I present in this section are closely related to the Weaknesses section where I share my thoughts, so please review both parts.

 1) In lines 201-2023, you mention that the responses generated by the LLM in the first prompt were quite long and easily detected and that you later adjusted the length. Have you tried examining the correlation between person A's ground truth responses and A-clone's responses? For example, you could use metrics such as ROUGE-1, ROUGE-L, Persona F1, and Win rate, as suggested in the paper you referenced, "Kai Zhang, Lizhi Qing, Yangyang Kang, and Xiaozhong Liu. Personalized LLM Response Generation with Parameterized Memory Injection" (2024).

  2) In lines 198-200, you state that, based on the questions you provided to A and A-clone, the test volunteers (such as A's family) had to predict which response belonged to A. Why didn’t you ask the test volunteers to come up with a set of questions they wanted to test, and then have both A and A-clone respond to those questions, incorporating them into the evaluation process? For example, in the paper you referenced, "Daniel Jannai, Amos Meron, Barak Lenz, Yoav Levine, and Yoav Shoham. Human or Not? A Gamified Approach to the Turing Test" (2023), they describe strategies that helped participants identify whether they were interacting with a chatbot or a human. Having such information for your model would provide deeper insights into its abilities.

  3) In lines 203-208, you mention that you applied an SVM to determine if person A’s responses were easily recognized based only on length, with an accuracy of 0.48. Have you considered using a state-of-the-art classification model? In my opinion, this accuracy doesn’t provide much information, given that you used a simple classifier like SVM.

  4) In line 770, regarding Table 2, you present some statistics related to the experiment. Did you perform any correlation tests to identify trends between the variables (e.g., gender, age range, etc.) and the test volunteers’ responses? For instance, it’s possible that individuals with a PhD level of education might be better at discerning whether something was generated by an AI.

**Details Of Ethics Concerns:**

I would like to thank the authors for mentioning the potential for undesirable conduct in the discussion. A tool like this can be useful in many domains where personalized information is valuable. For example, in sales, people might interact with chatbots designed to convince them to make a purchase. As noted in the paper, if this tool is used to create chatbots impersonating public figures, it could be exploited to scam individuals in various ways, such as through phishing emails. Generally, a personalized LLM is highly likely to introduce bias in the model's outputs. Therefore, it is crucial to implement fairness and bias detection measures to mitigate biased outcomes. This is something I believe the authors should address in their paper, and ideally, they should incorporate it into their experimental process. Another ethical consideration, which I have already mentioned in the section on weaknesses, is the omission of the concern raised by person A, who allowed the authors to use their data.

---

> ### Author Response · Authors · 2024-11-28
> **Part 1**
>
> We thank the reviewer for their detailed and insightful comments. The point-to-point answer is provided below.
>
> **Reply to Weaknesses**
>
> **A1)** We now explicitly state in both the abstract and the introduction that the data were in English.
>
> **A2.1)** We have modified those parts by using the LLM abbreviation.
>
> **A2.2)** We have modified those parts by using the SFT abbreviation.
>
> **A2.3)** We have carefully reviewed and corrected the spacing issues for the references throughout the manuscript.
>
> **A3)** Thank you for your suggestion. We have added a reproducibility statement at the end of the main paper, located before the references section on page 11.
>
> **B1)** We have revised the manuscript to explicitly state that A's explicit permission was obtained prior to collecting and using their personal data. This clarification has been added to the introduction part (line 050)
>
> **B2)** Thank you for your valuable feedback. We fully acknowledge this limitation and agree that expanding the experiment to include multiple users would strengthen the robustness and generalizability of our approach. We have now added this point in the Discussion Section (line 485).
>
> Creating an LLM-clone involves significant effort, including finding individuals who have stored a substantial amount of data for training, extensive data curation, large-scale LLM fine-tuning. Next, we need to find a large group of people with sufficient familiarity with the individual for the Turing-like tests. Thus, this effort is different from other experiments like assessing visual recognition which can be conducted on large numbers of individuals on mechanical turk and similar platforms. We are currently working on creating additional clones.
>
> That said, our focus here was not on developing a broadly applicable methodology for cloning but rather on addressing the feasibility, safety, privacy, and societal concerns raised by such a proof-of-concept LLM-clone. This is why we have submitted our work to the "alignment, fairness, safety, privacy, and societal considerations" track. We highlight in line 025, line 476, line 529 that this work is intended as a proof-of-principle.
>
> **B3)** Thank you for your feedback on this critical point. However, we were probably not clear and there may be a misunderstanding. In the "Related Work" section noted by the reviewer, the statement, "additionally, the datasets used for personalization tasks are often anonymized, making it difficult to trace their origins and perform thorough, systematic evaluations," refers to existing public datasets where the identities of the dataset owners—such as the individuals who wrote the emails—are anonymized. This statement is not about our work.
>
> Regarding anonymizing our dataset, we would like to clarify that we have already implemented the necessary PII (Personally Identifiable Information) anonymization during the data curation process before training.
>
> Perhaps the following point is obvious to the reviewer but all the participants knew that this was a test about person “A”. This was the whole point of the Turing-like tests where participants were asked whether a given answer came from “A” or “not A” or were provided multiple answers and had to select the one from “A”. Everything else was anonymized.  We used the term “A” throughout to keep the ICLR submission anonymous but we used “A”’s actual name in the test and we will change this in the final version of the manuscript.
>
> We have also updated the Related Work section (line 107) to eliminate any potential confusion.

---

> ### Author Response · Authors · 2024-11-28
> **Part 2**
>
> **B4)** Thank you for your valuable feedback and insightful suggestions.
>
> We have expanded our discussion to provide a more detailed overview of the question sources, topics, and their distribution, as well as the performance of participants during the Turing Test across two distinct subgroups: question source and question topics. These details, along with a subgroup comparison that yielded some interesting insights, have been included in Appendix A.8.
>
> To ensure that the LLM's performance is not due to overfitting on training data, we conducted a textual similarity analysis between our training set and test sets using ROUGE-1 and ROUGE-L metrics. Our results demonstrate that the distributions of the training and test sets are significantly different, suggesting that overfitting is unlikely. Details of this analysis have been added to Appendix A.6.
>
> **B5.1)** Thank you for the suggestion. We have incorporated a new paragraph in the Related Work section titled "Turing Test."(lines 113-121).
>
>
> **B5.2)** Thank you for the suggestion. We have already discussed several applications of personalized LLMs in the Related Work section (lines 100–102). We have refined this section by including citations for the papers suggested by the reviewer. However, due to page limitations, we did not expand this into a new paragraph. Additionally, we have updated the introduction to highlight potential applications of our methods (lines 064–068).
>
> **B5.3)** Thanks for recommending these amazing works on personalized LLMs. We now added these works in the related works (lines 100-102).
>
> **B5.4)** Thank you for the suggestion. We have already addressed the most relevant works in the Introduction section (lines 44–48) and outlined how our approach differs by addressing the limitations of prior work. Additionally, we have highlighted our main contributions at the end of the Introduction. Due to space constraints, we are unable to further expand this section.
>
> **Reply to Questions**
>
> **1** Thank you for suggesting these metrics. In the modified paper, we have incorporated ROUGE-1 and ROUGE-L as new metrics to compare the textual similarities between 701 responses generated by different models, providing an additional layer of evaluation alongside the Turing Test. The results demonstrate that A-clone outperforms the other LLM models, achieving the highest scores in both ROUGE-1 and ROUGE-L. (Check Appendix A.6 for details.)
>
> Also, we agree that using a Language Model as a judge and calculate Win Rate is an interesting and relevant approach, as recent research has shown LLMs have strong alignment with human preferences. This aligns with our planned future work, where we aim to implement such metrics to further validate our findings.
>
> **2** Thank you for the suggestion.  In fact, this approach aligns precisely with part of what we implemented during the Turing Test question dataset collection. Participants were asked to generate at least 10 questions they wished to ask the clone, resulting in 217 out of the 701 questions originating directly from participants. We now present results categorized by question types in Appendix A.8, and explicitly describe in the main text that some of the questions were participant-generated. (line 155) For further analysis, please refer to Appendix A.8.
>
> **3** Thank you for the feedback. The purpose of using an SVM classifier in the section referred to by the reviewer  was to assess whether response length alone could serve as a predictive factor for distinguishing AI-generated answers. This was motivated by the hypothesis that one possible strategy during the Turing Test could involve evaluating choices simply based on their length. Given that this was a very simple classification task and that there is a single scalar value for each response (its length), we did not seek to use more sophisticated classification models.
>
> **4** Thank you for your question.  In our paper, we discussed the potential correlation between *performance* and both *familiarity with A* and *confidence levels in Turing Test*. Beyond these factors, we have not identified any correlation between performance and other demographic variables (*age, gender, etc*). In the appendix A.9, we include a plot that examines the relationship between performance and *education level*, addressing the hypothesis raised by the reviewer. As illustrated in figure 17 and figure 18, no significant correlation can be observed.

---

> ### Author Response · Authors · 2024-11-28
> **Part 3**
>
> **Reply to Details Of Ethics Concerns**:
>
> Thank you for your acknowledgement and valuable feedback. Indeed, the purpose of our paper is to investigate the feasibility of creating an LLM-clone and to raise awareness of the associated concerns within the community. We have added a dedicated section discussing the ethical implications of our work in section 6 -- Ethical Considerations.
>
> Regarding your mention of "fairness and bias detection," we are not entirely certain of the specific context you are referring to. Our primary evaluation focuses on whether the clone can accurately mimic an individual's language behavior. In this context, biases and preferences—such as favoring ice cream over broccoli in one’s diet—are inherent to the individual being mimicked. These are intentional and form an essential part of the model's objective to authentically replicate the individual's behavior.
>
> That said, we agree that examining potential bias tendencies in such a model would be a valuable avenue for future research. In fact, we have conducted a moral reasoning test (Defining Issues Test) as one of the ten personality assessments. The results indicate that our clone is slightly more aligned with A’s personal moral choice tendencies compared to other LLM baselines. We acknowledge the importance of expanding such evaluations and plan to explore these aspects further in future work.
>
> We also would like to emphasize that our study received approval from the Institutional Review Board (IRB). Additionally, Person A provided informed consent for the use of their data in this research.
>
> We do acknowledge that there is much more to discuss about the ethics of personalized clones than the limited number of pages allows for.

---

### Author Response · Authors · 2024-11-28
**Global Reply to All Reviewers**

We would like to thank all the reviewers for their valuable questions and comments. In response, we have submitted a revised PDF of our paper, with all modifications highlighted in red in the main text. Below is a summary of the key changes made:

## Main Text Revisions

1. **General Improvements**:
   - Corrected typos, resolved inconsistencies, and added necessary details to enhance readability.

2. **Introduction Section**:
   - Emphasized the broad application potential of our work. (lines 64-67)

3. **Related Work Section**:
   - Cited the papers suggested by reviewers and added a new subsection, *Turing Test*, under Related Work. (lines 113-121)

4. **Personality Test Analysis**:
   - Expanded the discussion and provided an in-depth analysis of personality test results. (lines 426-472)

5. **Discussion Section**:
   - Highlighted the limitation of cloning only a single individual. (line 485)

6. **Ethical Considerations Section**:
   - Added a dedicated section discussing the ethical implications of our work. (lines 512-524)

7. **Reproducibility Statement**:
   - Included a reproducibility statement on page 11, before the References section. (lines 541-550)

## Appendix Expansion

We have significantly expanded the appendix to include additional results and analyses:

1. **Turing Test 1 Analysis (Appendix A.8)**:
   - Added a detailed introduction to the test questions and an in-depth analysis of results categorized by question types and sources.
   - Provided insights from these results that may guide future algorithm development and question design strategies for detecting LLM clones.

2. **Training Set Overview (Appendix A.7)**:
   - Included visualizations such as a word cloud and word-length distributions to provide a more comprehensive overview of the training set.

3. **Textual Similarity Tests (Appendix A.6)**:
   - Conducted two textual similarity analyses using ROUGE-1 and ROUGE-L metrics. Our findings indicate:
     - The training and test sets have distinct distributions, reducing the likelihood of overfitting.
     - The A-clone models exhibit greater lexical similarity to A’s ground truth answers compared to other LLMs.

4. **Reviewer-Requested Figures**:
   - A non-normalized confusion matrix **(Appendix A.5)**.
   - A correlation plot between participants’ performance and their educational levels **(Appendix A.9)**, showing no observed correlation. This aligns with the lack of correlations found for other demographic variables such as age and gender.

## Note on Page Limitations

Due to the page limit, we were unable to integrate all the new experiments into the main manuscript and included them in the appendix instead. If our paper is accepted, we plan to reorganize the manuscript to incorporate key findings—such as detailed subgroup Turing Test 1 results and textual similarity tests—into the main body.

## Note on Turing Test for Reviewers

We also would like to encourage reviewers to participate in the Turing Test we have prepared for reviewers, which contains 12 questions in total with real time feedback at the end of the test, and will provide a clearer understanding of the test questions. The link is included in the supplementary Google Drive folder at the end of the abstract, but here it is for your convenience: [link](https://illinois.qualtrics.com/jfe/form/SV_3sGqgw7SP8diwF8).

---

### Meta-Review · Area_Chair_8AUP · 2024-12-23

**Metareview:**

This is an interesting paper comparing responses from (i) a model fine-tuned on two decades of private data from a single individual, A to (ii) responses from A themselves to (iii) responses from non-A-related LLMs to (iv) responses from A’s human family members prompted to impersonate A.  Tests including “Turing-like” tests along with personality-based ones (e.g. MBTI, political, etc).  Reviewers appreciated the human subjects evaluation part of this paper, but flagged a number of concerns around the way human Q&A was generated (point 3 by 1HYm), some ethical considerations (many reviewers), generalization and reproducibility (partially addressed by the authors - we realize this is a private dataset of extremely personal data).  This is an interesting borderline paper.

**Additional Comments On Reviewer Discussion:**

Those reviewers who responded to the comprehensive rebuttal weakly increased either their opinion of the paper or, indeed, their scores themselves - yet even both of those reviewers did still find unresolved weaknesses in the work.  All reviewers acknowledge that this type of human subjects eval is complex.

---

### Decision · Program_Chairs · 2025-01-22

Reject